# Bootstrap simulations for evaluating the model estimation of the extent of cross-pollination in maize at the field-scale level

Bo-Jein Kuo[1,2,3], Yun-Syuan Jhong[4], Tien-Joung Yiu[5], Yuan-Chih Su[1], Wen-Shin Lin[6]*

1 Department of Agronomy, National Chung Hsing University, Taichung, Taiwan, 2 Innovation and Development Center of Sustainable Agriculture (IDCSA), National Chung Hsing University, Taichung, Taiwan, 3 Pervasive AI Research (PAIR) Labs, Hsinchu, Taiwan, 4 Central Region Branch, Agriculture and Food Agency, Council of Agriculture, Executive Yuan, Taiwan, 5 Tainan District Agricultural Research and Extension Station, Council of Agriculture, Executive Yuan, Taiwan, 6 Department of Plant Industry, National Pingtung University of Science and Technology, Pingtung, Taiwan

* wslin@mail.npust.edu.tw

**Data Availability Statement:** The original CP rate (%) of Table 1 in various rows and the distances from the pollen source in the 2009-1, 2009-2, and

## Abstract

With the recent advent of genetic engineering, numerous genetically modified (GM) crops have been developed, and field planting has been initiated. In open-environment cultivation, the cross-pollination (CP) of GM crops with wild relatives, conventional crops, and organic crops can occur. This exchange of genetic material results in the gene flow phenomenon. Consequently, studies of gene flow among GM crops have primarily focused on the extent of CP between the pollen source plot and the adjacent recipient field. In the present study, Black Pearl Waxy Corn (a variety of purple glutinous maize) was used to simulate a GM-maize pollen source. The pollen recipient was Tainan No. 23 Corn (a variety of white gluti-nous maize). The CP rate (%) was calculated according to the xenia effect on kernel color. We assessed the suitability of common empirical models of pollen-mediated gene flow (PMGF) for GM maize, and the field border (FB) effect of the model was considered for small-scale farming systems in Asia. Field-scale data were used to construct an optimal model for maize PMGF in the maize-producing areas of Chiayi County, southern Taiwan (R. O.C). Moreover, each model was verified through simulation and by using the 95% percentile bootstrap confidence interval length. According to the results, a model incorporating both the distance from the source and the FB can have optimal fitting and predictive abilities.

## Introduction

With improvements in biotechnology and genetic engineering, the area assigned to the cultivation of genetically modified (GM) crops has increased by 1.9 million hectares from 2017 to 2018 [1]. Among the 26 countries growing GM crops in 2018, five countries grew 91% of these crops [1]. Moreover, more than 70% of plants produce offspring through cross-pollination (CP) between species; this is a commonly observed evolutionary phenomenon in plants, and pollen is the primary medium for this process [2]. Therefore, when GM crops are cultivated in

2010-1 fields are available on Figshare (https://doi.org/10.6084/m9.figshare.13370756.v1) where distance= distance from the pollen source (m), CP= cross-pollination rate, Po= the CP rate (%) at the edge of the pollen recipient field, and ID= isolation distance.

**Funding:** The Ministry of Science and Technology provided funding for the publication fee in the form of a grant awarded to BJK, through Pervasive AI Research (PAIR) Labs, Taiwan (108-2634-F-005-003), and the Innovation and Development Center of Sustainable Agriculture, from The Featured Areas Research Center Program, within the framework of the Higher Education Sprout Project by the Ministry of Education (MOE) in Taiwan. No additional external funding was received for this study. The funders had no role in study design, data collection and analysis, decision to publish, or preparation of the manuscript.

**Competing interests:** The authors have declared that no competing interests exist.

an open environment, their pollens can spread to distant locations through airflow [3]. CP can occur from GM crops to wild relatives, conventional crops, and organic crops through pollen dispersal. This leads to an exchange of genetic material, and the resulting gene flow phenomenon affects neighboring crops. Pollen-mediated gene flow (PMGF) can be assessed by evaluating the CP rate (%) from the pollen source plots to recipient fields.

Maize (*Zea mays* L.) is a typical wind-pollinated crop. Pollen shedding begins before the tassel completely appears and lasts for approximately 5 to 6 days. Normally, the tasseling period is out of phase with the silking period in maize, resulting in a high rate of CP. However, the probability of self-pollination remains approximately 5% [4, 5]

Generally, for wind-pollinated species, pollen flow and CP decrease with distance from the pollen source, and this decrease follows a leptokurtic distribution. The length and shape of the distribution's tail depend on biological and environmental factors [6, 7]. Because maize pollen grains are heavier (approximately 0.25 μg) and larger (diameter approximately 70–100 μm) than those of other wind-pollinated species, maize pollen is typically spread over a limited distance. Therefore, the CP rate (%) decreases rapidly with distance, and the distribution of this reduction often has a fat tail. Thus, relative to other distributions, a leptokurtic distribution better models the relationship between CP rate (%) and distance from the maize pollen source [7].

To prevent the pollution of non-GM crops or related derivative products through unintentional CP and to ensure consumers' freedom of choice between GM and non-GM products, the European Union (EU) set the tolerance threshold for the adventitious presence (AP) of GM events at 0.9% (Regulation EC No. 1829/2003). Isolation distance (ID) is a common coexistence measure used to minimize the AP. Additionally, under current EU regulations, member states are responsible for establishing individual coexistence laws. However, these states have substantial discrepancies between their regulations governing the required and recognized IDs for planting GM maize [8]. The differences between the required and regulated IDs for maize in EU member states in the EU are fairly large [8], ranging from an ID regulation of 15 m in Sweden to that of 800 m in Luxemburg for forage corn [9, 10]. Therefore, a clear understanding of the PMGF distribution is required to determine a suitable ID for actual conditions.

Studies have modeled the PMGF for maize. Bannert and Stamp [11] investigated the CP trend following the long-distance dispersal of maize pollen. The maize fields had areas of 0.5–1.5 ha, and the distance between the source and recipient fields was 50–4500 m. The results indicated that vertical wind and gusts resulting from thermal or turbulence effects were the primary causes of long-distance CP. In addition, the average CP rates (%) of the recipient fields were >0.02%. Ma et al. 2004 [4] categorized field experimental data into downwind and upwind subsets and used exponential equations to investigate the relationship between the CP rate (%) of the conventional maize hybrid and the distance from the neighboring GM-maize fields. The results indicated a CP rate (%) of 0% at distances of at least 30 m from the pollen source. Moreover, to determine an effective isolation buffer between the pollen source and recipient fields and maintain an average CP rate (%) less than the 0.9% EU threshold, Gustafson et al. 2006 [12] used log decay functions to construct an empirical model. The simulation results indicated that when the source field covered an area of 0.07–0.65 ha, the average CP rate (%) of the recipient 1 ha field was less than 0.9%, with an ID of 20 m and border rows. In 2008, Della Porta et al. [5] used downwind and upwind experimental data to compare the fitting abilities of the following three empirical models to describe the relationship between CP rate (%) and distance from the pollen source: the exponential model, the log–log model, and the log–square model. The results indicated that the log–log model (a power law equation) had the best fit. However, the CP rate (%) was overestimated for distances within 3 m from the

recipient field to the pollen source when using this model. Šuštar-Vozlič et al. 2010 [13] used inverse power functions to develop a suitable CP model. Numerous samples must be collected and investigated to accurately estimate the CP of maize fields [13]. However, if an appropriate model and set of sampling methods are employed to calculate CP in the field, the required sample size can be reduced.

To perform farm-scale evaluation of maize fields with an area less than 5 ha, Weekes et al. [14] used experimental data collected from 55 experimental sites in England and developed a gene flow model. This model comprised two stages: (1) second-order log equations were adopted to determine the probability that a sample has a GM content of 0% at a given distance within the recipient field and (2) a beta distribution was used to calculate the mean proportion of GM content at a given distance. Based on this model, a separation distance of 3 m was recommended to ensure that the neighboring crop had a CP rate (%) less than the 0.9% threshold when using a square source field with an area of $150 \times 150$ m$^2$.

Most PMGF studies of GM maize have been conducted in EU member states or North America on large-scale fields. However, PMGF results might be affected by environmental factors, particularly regional climate, agricultural landscape, and experimental materials. Therefore, real field experiments on PMGF from pollen source plots to recipient fields in a small-scale farming system are warranted.

In this study, a farm environment with both GM and non-GM crops was designed to investigate the gene flow of GM crops and subsequently reduce the risk of GM gene flow to conventional crops. Although many countries have permitted GM crops to be commercially cultivated over large areas, no GM crops have been approved for commercially cultivation in Taiwan. Moreover, in the small-scale agricultural landscapes of Taiwan, crop fields are often separated by field borders (FBs), such as roadways between fields. Therefore, the influence of FBs on CP was also considered. This is the first comprehensive investigation of the effects of ID and FBs on the CP rate (%) of maize in Taiwan, an island with a subtropical climate.

Chiayi County is a primary maize-producing area of Taiwan. Therefore, experiments were conducted in Puzih City, Chiayi County (23°47′N, 120°26′E) during the growing seasons of 2009 and 2010 to determine the optimal model for CP. The empirical models of pollen flow used in other studies were referenced to develop a CP model for maize production areas with small-scale agricultural landscapes. Moreover, the fitting abilities of the empirical models were compared, and the fitting stability and predictive abilities of the optimal model were evaluated using 1,000 bootstrap simulations. The results can aid governments in establishing coexistence systems for GM and non-GM crops in field allocation for the cultivation of GM maize. This information can also be a reference for other Asian counties with similar farming systems, such as Japan, Korea, and the Philippines.

## Materials and methods

### Field design

In this study, experiments were conducted at the Puzih Branch Station of the Tainan District Agricultural Improvement Station from 2009 to 2010. Moreover, FBs are common in the agro-ecosystems of Taiwan because fields are often separated by roadways. Therefore, to establish a model that includes both the distance from the source and the FB, the following three field experiments were conducted (Fig 1): (1) During the first crop season in 2009 (2009–1), a study was designed to investigate circumstances in which the pollen source neighbors the pollen recipient field. Because the prevailing wind during 2009–1 is from the south, the pollen source was designated at the southern edge, and the pollen recipient was downwind from the pollen source (Fig 1A). The total site area was approximately 0.42 ha ($84.5 \times 50$ m$^2$), and the pollen

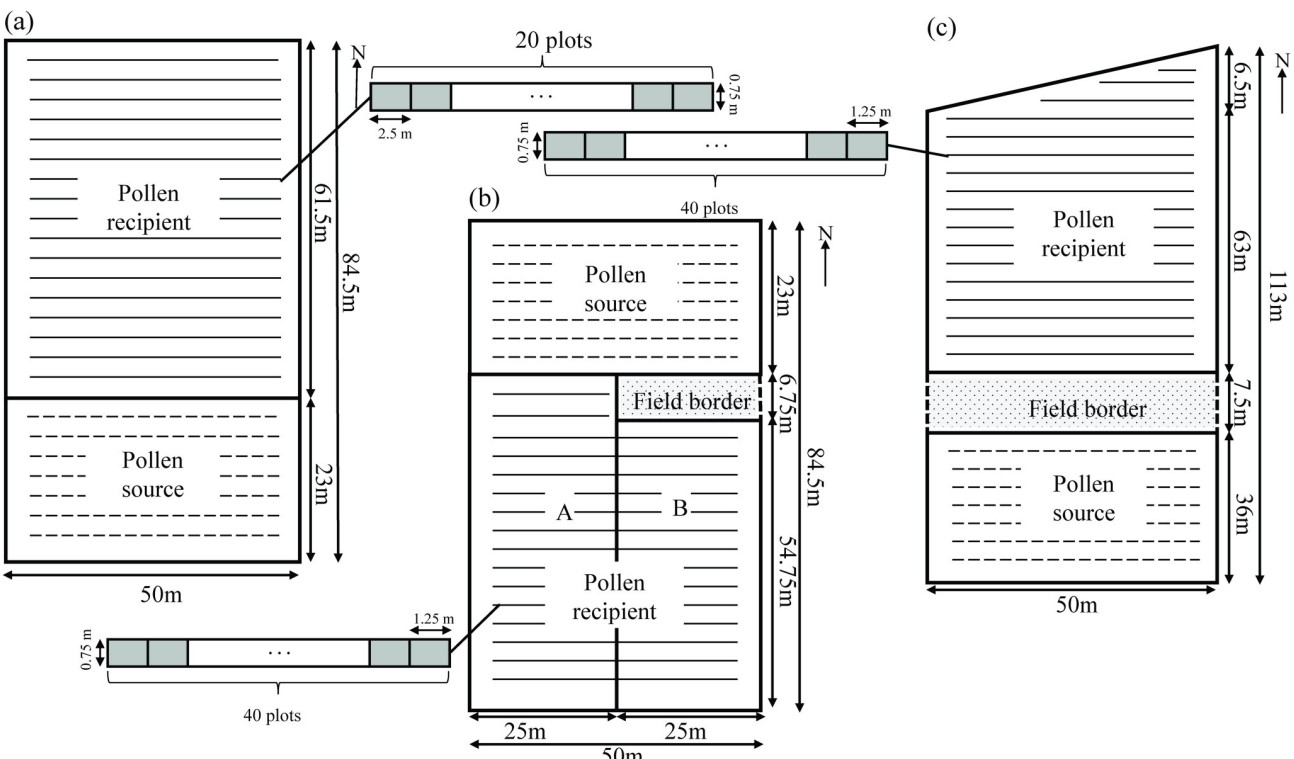

**Fig 1. Field design and sampling plots layout for the experiments.** Crop seasons (a) 2009–1, (b) 2009–2, and (c) 2010–1. The solid line indicates the pollen recipient, the dashed line indicates the pollen source, and the network node indicates the FB. The example of sampling plots layout is illustrated in the first row of each experiment.

source to recipient field area ratio was approximately 1:2.6. (2) During the second crop season in 2009 (2009–2), a study was designed to evaluate the effects of the FB. The pollen recipient field was divided into part A (without an FB; 2009-2A) and part B (with an FB; 2009-2B). Part A neighbored the pollen source, and part B had an FB with a width of 6.75 m to provide separation from the pollen source. Because the prevailing wind during fall is from the north, the pollen source was designated at the northern edge (Fig 1B). The total site area was approximately 0.42 ha ($84.5 \times 50$ m²), and the pollen source to recipient field area ratio was approximately 1:2.5. (3) During the first crop season in 2010 (2010–1), an experiment was conducted with an FB larger than that in 2009-2B. Maize fields were separated with an FB with a width of 7.5 m between the recipient field and the pollen source (Fig 1C). Because the prevailing wind during summer is from the south, the pollen source was designated at the southern edge. The total site area was approximately 0.55 ha, and the pollen source to recipient field area ratio was approximately 1:1.8.

## Plant culture

For this experiment, two commercial glutinous maize varieties with different grain colors were selected from those grown in Taiwan. Black Pearl waxy corn (Known-You Seed Co., Kaohsiung, Taiwan), which has purple grains, was used as the pollen source. The period from planting to flowering of this variety is approximately 40–50 days. Moreover, Tainan No. 23 corn, which has white grain and is suitable for planting in central and southern Taiwan, was used as the pollen recipient. The kernel pericarp color of the maize resulting from the xenia effect (i.e., purple grains on the white ears of the pollen recipient) was used to determine the CP rate (%).

The xenia effect of the maize is caused by the effect of the different pollen source gene resulted in endosperms on the development of the seeds.

We adopted conventional farming management methods in this study. The distance between individual plants in a row was 0.25 m, and the distance between rows was 0.75 m. Maize fields were planted with densities of 53,018 plants ha$^{-1}$ in the 2009–1 field, 53,251 plants ha$^{-1}$ in the 2009–2 field, and 53,467 plants ha$^{-1}$ in the 2010–1 field (Fig 1).

## Plant dates

To ensure congruence between releases of male pollen from the purple maize and the silking period of the white maize, the purple maize was sown in two batches according to planting time. The white maize variety flowers slightly later than the purple variety does in Puzih City, Chiayi County. Therefore, for the 2009–1 experiment, the white maize was sown on 8 May, the first batch of purple maize was sown on 11 May, and the second batch of purple maize was sown on 13 May. However, the results of the 2009–1 experiment indicated no clear difference between the flowering periods of the white and purple maize varieties. Therefore, in subsequent experiments, the first batch of purple maize was sown 3 to 5 days before the white maize was sown, and the second batch of the purple maize was sown at the same time as the white maize was. In the 2009–2 experiment, the white maize was sown on 16 October, the first batch of purple maize was sown on 13 October, and the second batch of purple maize was sown on 16 October. In the 2010–1 experiment, the white maize was sown on 3 May, the first batch of purple maize was sown on 28 April, and the second batch of purple maize was planted on 3 May.

## Climate monitoring during the flowering period

Meteorological information during the experiments, including wind speed and wind direction, was measured at the Central Weather Bureau's Yichu Branch Station (23˚36'N, 120˚28'E). Wind speed and wind direction were recorded hourly from 6 a.m. to 4 p.m. for 7 days before and after the silking period of the pollen recipient. Changes in wind rose plots were used to interpret trends in wind speed and direction.

## Data collection and CP (%) rate calculation

In this study, to explore the relationship between the CP rate (%) and the distance from the pollen source, ears were measured for each small plot by using the census method. The 2009–1 field was investigated using a census scale with small sample plots with areas of $2.5 \times 0.75$ m$^2$ (number of missing plots: 5; number of measured plots: 1,635). Furthermore, to identify and precisely describe the gene flow trends in maize, ears were measured with small sample plots with areas of $1.25 \times 0.75$ m$^2$ in the 2009–2 (number of missing plots: 10, number of measured plots: 3,110) and 2010–1 fields (number of missing plots: 392, number of measured plots: 3,248).

Visual inspection of the ears of the pollen recipient and xenia counting were used to calculate the CP rate (%). That is, the CP rate (%) was evaluated by counting the number of purple xenia kernels on the white pollen recipient according to the following formula (Eq 1) [5, 11]:

$$\mathrm{CP}(\%) = \left[\sum\nolimits_{i=1}^{n} \mathrm{Ear}_i / (n \times \mathrm{AVK})\right] \tag{1}$$

where $n$ is the number of ears in the sample plot, Ear$_i$ is the number of purple kernels on the $i$th ear in the sample plot, and AVK is the average number of kernels for an ear in the field. To determine AVK, two ears were randomly chosen from each sampling plot, and the total

number of kernels was counted for each selected maize ear to calculate the average number of kernels for an ear.

## PMFG models

In previous studies, the majority of models describing the relationship between the CP rate (%) and the distance from the source pollen can be categorized into the following types: the exponential model, log–log model, and log–square model. In these equations, distance refers to the distance (m) from the sampled ears to the edge of the pollen source field, and $a$ and $b$ are model parameters [4, 5].

To consider the effects of FBs, the field designs of the 2009–2 and 2010–1 experiments included FBs. Therefore, in addition to employing the aforementioned models, the empirical modeling approach proposed by Gustafson et al. [12] was investigated. This model contained information for the adventitious presence of the GM trait in seeds within the recipient field (%) (AP), the fraction of pollen containing the GM trait ($F_{GM}$), the CP rate in the row closest to the edge of the pollen recipient field ($P_0$), the ID from the edge of the source field to the recipient field (ID; this was the same as FB in the present study), distance from edge of recipient field nearest to the source ($x$), and the width of the border rows (in meters) for the non-GM crop planted between the source and recipient fields (BR) (Eq 2).

$$CP = AP + (F_{GM}P_0 10^{-[0.1\sqrt{ID}+0.2\sqrt{x+BR}]})$$                   (2)

After individual field trial data were fit using the empirical modeling approach, the trend of the gene flow within recipient or border rows could be reduced, as indicated by the coefficient of the proportionality constant, and twice as effective as the ID (the unplanted isolation buffer), as reported in Gustafson et al. [12]. Consequently, in this study, the proportionality constants for FB and the distance within the recipient field were re-estimated based on the data from field experiments conducted during the growing seasons of 2009 and 2010.

Furthermore, in the European commercial biotech environment, because the AP of the unavoidable GM traits of seeds must be less than 0.3%, the AP was set at 0.003 in the model [12]. However, under the current planting regulations for GM crops in Taiwan, no GM crops have been approved for commercial cultivation in open fields. Moreover, planted seeds cannot be admixed with GM seeds. Therefore, the AP for the empirical modeling approach should be adjusted to 0% for the conditions of Taiwan's current agricultural environment. In addition, to conform to the planting regulations that forbid the cultivation of GM crops, $F_{GM}$ was set to 1; that is, all pollen grains containing GM traits introduced a risk of spreading GM traits through pollination [12]. Gustafson et al. [12] also indicated that the mathematical descriptions of the relationships between gene flow and distance from the pollen source are the same as those for the recipient and border rows. However, rows in the isolation field with non-GM maize between the source and recipient fields were not considered as border rows. Consequently, in the empirical modeling approach, BR was set to 0.

The models with the coefficient values estimated from the data and proposed by Gustafson et al. [12] were evaluated in this study. Furthermore, the models with and without 0.003 AP were investigated to assess the assumption of receptor seeds carry GM trait due to 0.3% AP. In summary, the following five models (Eqs 3–7) were used to describe the relationship between the CP rate (%) and the distance from the pollen source in the field experiments:

$$CP_1 = b \times e^{a(distance)}$$                   (3)

$$CP_2 = b \times distance^a$$                   (4)

$$CP_3 = b \times 10^{a\sqrt{\text{distance}}} \tag{5}$$

$$CP_4 = P_0 \times 10^{(a\sqrt{\text{FB}} + b\sqrt{\text{distance}})} \tag{6}$$

$$CP_5 = 0.003 + P_0 \times 10^{(a\sqrt{\text{FB}} + b\sqrt{\text{distance}})} \tag{7}$$

where $CP_i$ refers to the CP rate (%) estimated using the $i$th model, distance is the distance (m) from the sampled plot to the edge of the pollen source field, $FB$ is the width of the FB, which functions as an isolation buffer, $P_0$ is the parameter that describes CP rate (%) at the edge of the pollen recipient field, and $a$ and $b$ are model parameters.

## Simulation analysis

To evaluate the fitting performance and predictive ability of the models and the stabilities of the estimated parameters, bootstrapping was performed to simulate the population distribution. Bootstrapping is a nonparametric simulation method that uses a computationally intensive resampling technique to assess the variance of a statistic or parameter [15]. The benefit of the bootstrapping method is that the population distribution need not be assumed when generating data sets through random sampling with replacement.

In this study, random sampling with replacement was applied to the actual data set across lumped together across the three fields. A total of 8,005 observations of raw data were used to generate new data sets through 1,000 bootstraps. Then, the PMGF models were fitted according to the 1,000 bootstrap samples. To compare the stabilities in the fitting performance and predictive ability of the calibration and validation sets, confidence intervals were calculated for specific evaluation criteria with a 5% significance level on the basis of the percentile bootstrap (PB) confidence interval method [16]. By sorting the evaluation criteria of the 1,000 bootstrap samples from smallest to largest, the 95% PB confidence intervals were calculated between the 2.5th and 97.5th percentiles.

## Statistical analysis

In this study, Statistical Analysis System (SAS) version 9.4 (SAS Institute, Cary, NC, USA) was used to conduct the statistical analysis. The PROC NLMIXED was used for fitting the models. Before fitting the models, the CP rates were transformed to the count data, and the count data were assumed to follow a Poisson distribution. Thereafter, the collected experimental data were randomly partitioned into two sets, a calibration set (two thirds of the samples) and a validation set (one third of the samples), for fitting the models and validating their predictive abilities, respectively.

To compare the fitting performance of the models in describing the relationship between the CP rate (%) and distance from the pollen source, evaluation criteria were based on the deviance and Akaike information criterion (AIC) [17]. In addition, based on the validation data, deviance and AIC were used to evaluate the fitting abilities of the fitted models. Furthermore, the correlation coefficient (r) and scatter plots were used to assess the predictive ability of each fitted model based on the validation data.

Finally, similarly to the analysis of the collected experimental data, the simulated data for each bootstrap sample were also categorized into a calibration set (two thirds of the samples) and a validation set (one third of the samples) to fit the models and validate their predictive abilities. In the simulated data of each bootstrap sample, the values of deviance, AIC, and

parameters were estimated for each model. Then, standard deviations (SDs), confidence intervals of deviance and AIC were calculated to assess the stabilities of the performance of each model. For the simulated validation data, the correlation coefficient was also calculated to evaluate the predictive ability of each model.

## Results and discussion

### Weather patterns

The average wind speed and direction were monitored for 7 days before and after the 50% pollen silking of the pollen recipient. The average wind speed was higher and more varied during the 2009–1 and 2010–1 crop seasons than during the 2009–2 season (Fig 2). During the investigation period in the 2009–1 season (June 17 to July 1), the prevailing winds were mostly from S to SE (Fig 3A). The daily average wind speed was $4.57 \pm 2.06$ m s$^{-1}$ (Fig 2A). Because of the influence of the peripheral circulation of Typhoon Linfa during the flowering period (June 19–22), the gust speeds reached up to 22.3 m s$^{-1}$. In 2009–2, the prevailing wind was mostly from N to NNW. The daily average wind speed was $3.88 \pm 1.27$ m s$^{-1}$ during the investigation period (November 4–18; Fig 3B). The gust speeds were up to 13.4 m s$^{-1}$. In 2010–1, the daily average wind speed was $4.33 \pm 0.26$ m s$^{-1}$, and the gust speeds were up to 14.4 m s$^{-1}$. The prevailing winds were mostly from SSE during the investigation period (June14–28; Figs 2C and 3C).

### Gene flow trends

The CP rate (%) tended to decrease with increasing distance from the pollen source (Fig 4). Table 1 shows the average CP rate (%) for various rows and the distance from the pollen source in the 2009–1, 2009–2, and 2010–1 experiments.

The highest measured CPs rates (%) occurred in the rows of the recipient fields closest to the pollen source. The CP rate (%) was the highest (93.7%) in the 2009-2A experiment (without an FB). Moreover, the average CP rate (%) in the row closest to the pollen source was 74.29% in the 2009-2A experiment (without an FB), 36.12% in the 2009-2B experiment (with a 6.75 m FB), 27.58% in the 2010–1 experiment (with a 7.5 m FB), and 27.24% in the 2009–1 experiment (without an FB). These results indicated that FBs may enhance pollen exchange in the row of the pollen recipient closest to the pollen source. The average CP rate (%) of the other rows decreased as the distance from the pollen source increased (Table 1). As expected, fields neighboring the pollen source (e.g., 2009-2A) had higher CP rates (%) than those separated by FBs (e.g., 2009-2B and 2010–1).

In the 2009–1 experiment, the average CP rate (%) declined from 27.24% to 2.49% at a distance from the pollen source of approximately 9 m. In the 2009–2 field, the average CP rate (%) decreased from 74.29% to 2.49% at a distance from the pollen source of approximately 18 m. However, when the pollen recipient field was separated from the pollen source by an FB of 6.75 m, a rapid decline in the average CP rate (%) from 36.12% to 2.50% was observed at a distance from the FB of approximately 9 m. When the width of the FB was increased to 7.5 m in the 2010–1 field, the average CP rate (%) decreased from 27.58% to 2.18% at a distance from the FB of approximately 4.5 m. According to these results, the average CP rate (%) rapidly decreased with increasing distance from the pollen source and with increasing FB width. In addition, with the 6.75 m natural barrier in the 2009-2A field, the average CP rate (%) of the row closest to the pollen source was only 7.43%. Therefore, border rows may be more effective buffer zones relative to FBs for decreasing the CP rate (%), whereas FBs are more suited to long-distance pollination events. However, when the distance was increased to 18.75 m in the 2009–2 experiments, the average CP rates (%) of the rows in the 2009-2B field were mostly lower than those in the 2009-2A field (Table 1).

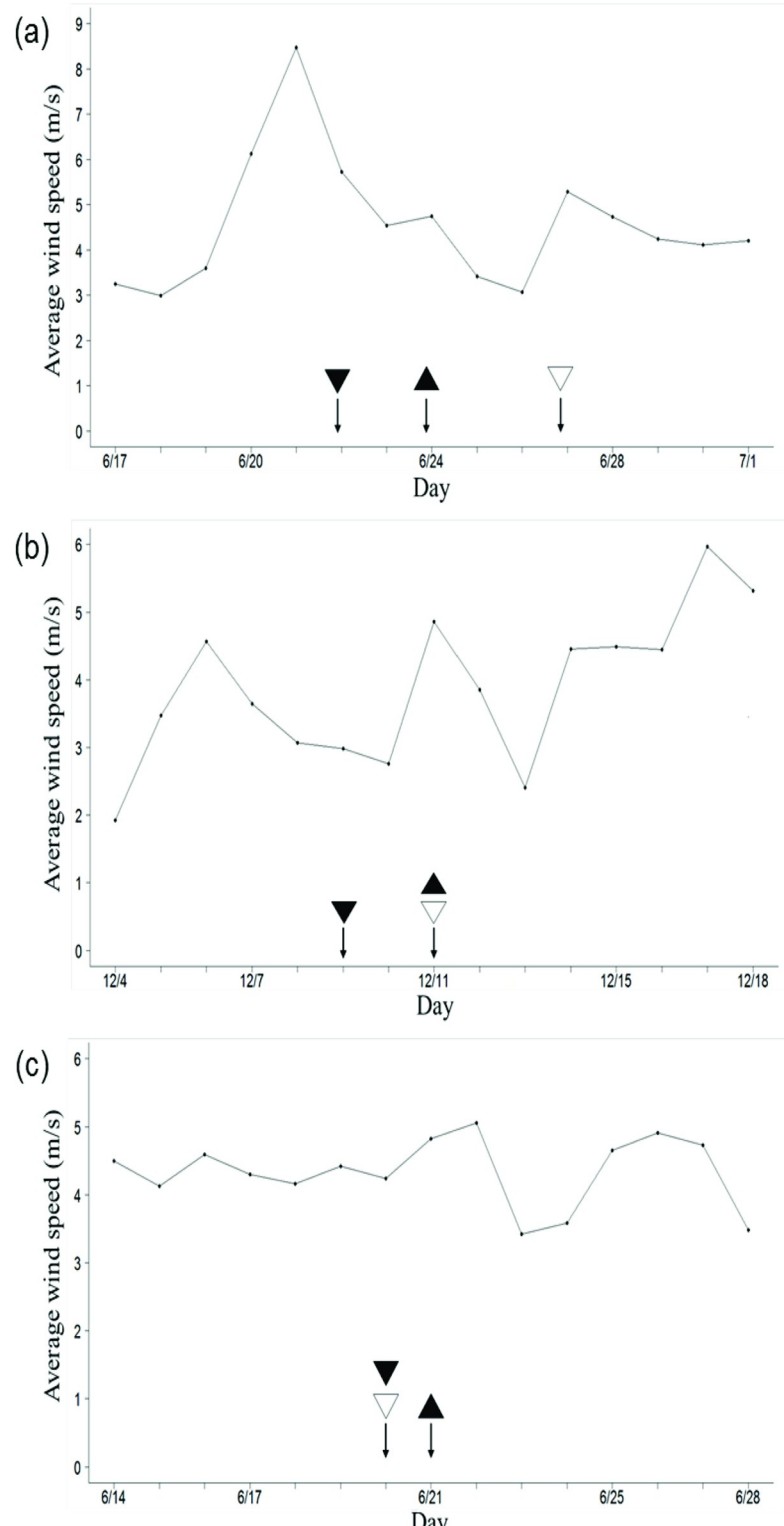

**Fig 2. Average wind speed (m/s) for 7 days before and after 50% pollen silking.** Crop seasons (a) 2009–1, (b) 2009–2, and (c) 2010–1. ▲: silking date of pollen recipient; ▼: shedding date of pollen recipient, and ▽: shedding date of pollen source.

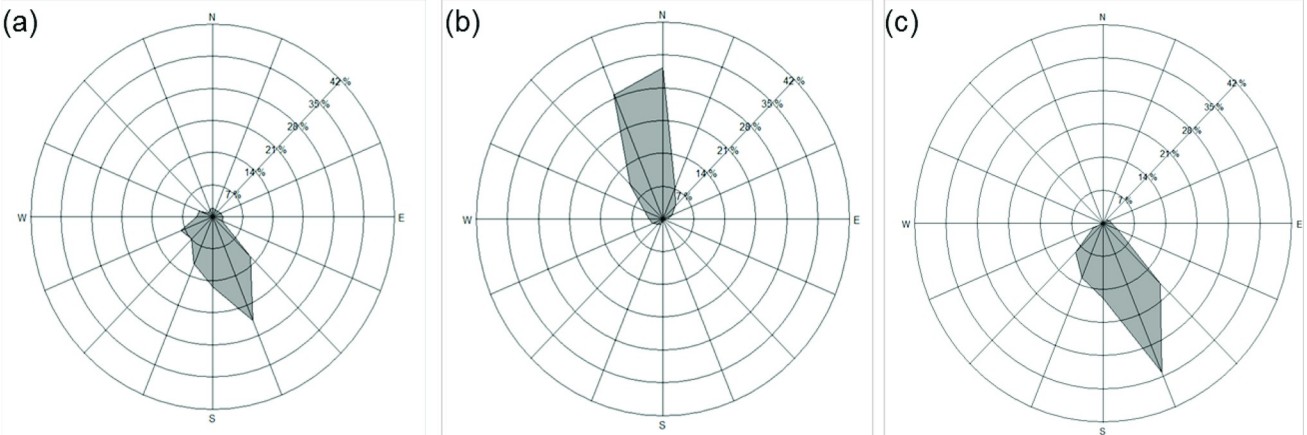

**Fig 3. Wind roses of wind direction frequencies measured hourly between 6:00 a.m. and 4:00 p.m..** (a) June 17 to July 1, 2009; (b) November 4–18, 2009; and (c) June 14–28, 2010. The scales measure the hourly occurrence gusts from different directions.

The 2009–1 field was influenced by the peripheral circulation of a typhoon during the flowering period. The sudden heavy rain and gusts resulted in the lodging of corn plants and a lower CP rate (%).

According to current regulations in Taiwan, the ID should result in a GM content of 0%. However, even at maximum distance from the source (i.e., 75 m), the average CP rate (%) remained as high as 0.1%. To meet the 0.9% EU threshold, the 2009-2A field (without an FB) required a distance of at least 39.75 m from the pollen source, and the 2009-2B field (with a 6.75 m FB) required a distance of approximately 36 m from the pollen source. Moreover, the 2010–1 field (with a 7.5 m FB) required a distance of approximately 18 m. Furthermore, for the 5% GM content threshold in Taiwan, the 2009-2A field (without an FB) required a distance of approximately 9.75 m from the pollen source, the 2009-2B field (with a 6.75 m FB) required a minimum distance of 12.75 m, and the 2010–1 field (with a 7.5 m FB) required a distance of approximately 9.75 m. Because the 2009–1 field was influenced by a typhoon during the flowering period, it is not included in this discussion.

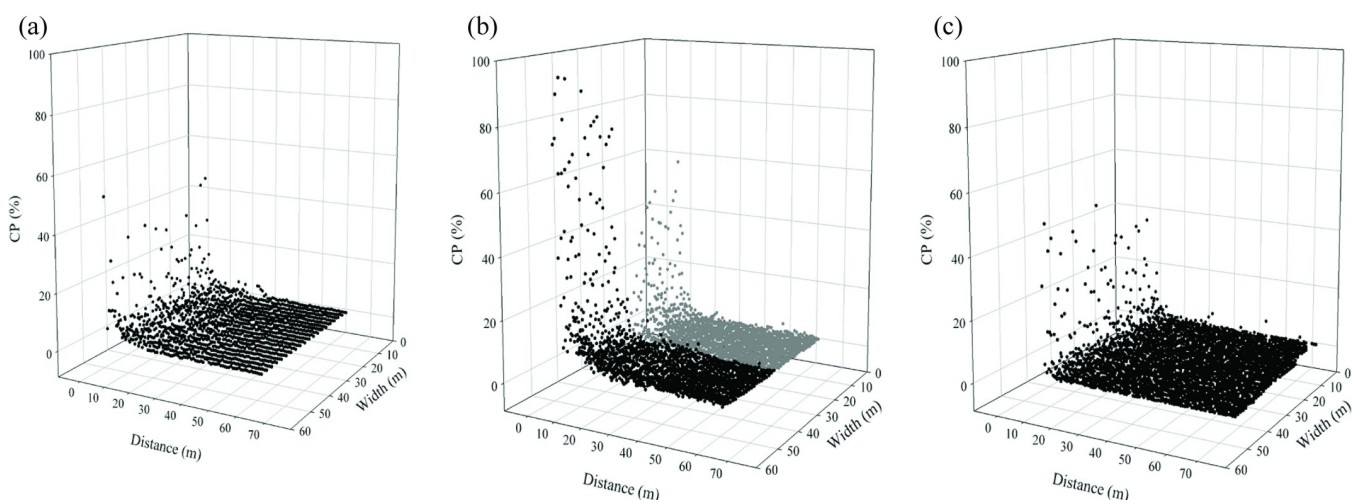

**Fig 4. Scatter plots of pollen CP in the maize fields.** Crop seasons (a) 2009–1, (b) 2009-2A (black dot) and 2009-2B (gray dot), and (c) 2010–1.

**Table 1. Average CP rate (%) in various rows and the distances from the pollen source in the 2009–1, 2009–2, and 2010–1 fields.**

| Site | 2009–1 | 2009–2 | | 2010–1 |
|---|---|---|---|---|
| | | A | B | B |
| Distance(m)[a] | without FB[b] | without FB | with 6.75 m FB | with 7.5 m FB |
| 0.75 | 27.24 ± 10.55[c] | 74.29 ± 10.70 | | |
| 2.25 | 8.85 ± 5.47 | 27.39 ± 9.75 | FB | |
| 3.75 | 4.51 ± 2.98 | 12.56 ± 5.69 | | FB |
| 5.25 | 2.85 ± 1.83 | 7.75 ± 4.95 | | |
| 6.75 | 2.88 ± 2.82 | 7.43 ± 5.97 | 36.12 ± 11.14 | |
| 7.50 | 3.25 ± 1.77 | 5.19 ± 4.07 | 20.08 ± 7.38 | 27.58 ± 11.44 |
| 9.75 | 2.49 ± 1.81 | 4.98 ± 4.23 | 8.01 ± 5.05 | 4.60 ± 6.23 |
| 12.75 | 1.17 ± 1.07 | 2.96 ± 2.27 | 4.04 ± 2.18 | 1.37 ± 1.33 |
| 15.75 | 1.25 ± 1.27 | 2.49 ± 1.57 | 2.50 ± 1.65 | 0.68 ± 0.88 |
| 18.75 | 0.52 ± 0.99 | 2.49 ± 1.92 | 1.49 ± 0.73 | 0.95 ± 1.13 |
| 21.75 | 1.00 ± 0.80 | 2.24 ± 1.33 | 1.61 ± 0.83 | 0.61 ± 0.53 |
| 24.75 | 0.50 ± 1.00 | 1.53 ± 0.88 | 1.53 ± 0.79 | 0.57 ± 0.87 |
| 27.75 | 0.51 ± 0.51 | 1.75 ± 1.02 | 1.11 ± 0.53 | 0.60 ± 0.71 |
| 30.75 | 0.25 ± 0.31 | 1.50 ± 0.99 | 1.16 ± 0.74 | 0.36 ± 0.60 |
| 33.75 | 0.29 ± 0.22 | 1.36 ± 0.96 | 1.09 ± 0.74 | 0.22 ± 0.41 |
| 36.75 | 0.19 ± 0.28 | 0.81 ± 0.76 | 0.81 ± 0.34 | 0.20 ± 0.40 |
| 39.75 | 0.15 ± 0.18 | 1.34 ± 1.93 | 0.57 ± 0.33 | 0.22 ± 0.39 |
| 42.75 | 0.18 ± 0.18 | 0.55 ± 0.39 | 0.80 ± 0.46 | 0.15 ± 0.40 |
| 45.75 | 0.20 ± 0.18 | 0.59 ± 0.35 | 0.47 ± 0.28 | 0.23 ± 0.38 |
| 48.75 | 0.20 ± 0.12 | 0.66 ± 0.50 | 0.38 ± 0.37 | 0.14 ± 0.37 |
| 52.50 | 0.19 ± 0.15 | 0.64 ± 0.46 | 0.40 ± 0.44 | 0.18 ± 0.20 |
| 56.25 | 0.18 ± 0.17 | 0.60 ± 0.56 | 0.49 ± 0.44 | 0.07 ± 0.11 |
| 60.00 | 0.11 ± 0.06 | 0.55 ± 0.38 | 0.22 ± 0.18 | 0.13 ± 0.35 |
| 63.75 | | | | 0.08 ± 0.09 |
| 67.50 | | | | 0.12 ± 0.24 |
| 71.25 | | | | 0.27 ± 0.64 |
| 75.00 | | | | 0.11 ± 0.17 |

[a]: distance from the pollen source (m).

[b]: FB indicates a field border between the pollen source and pollen recipient.

[c]: mean ± standard deviation.

## PMGF model

The estimated values of the regression parameters, the deviance, and AIC for each CP model based on the complete data are presented in Table 2. The estimated values of $a$ for $CP_1$, $CP_2$,

**Table 2. Regression parameters, deviance, and AIC for the complete dataset in each model.**

| Model | $P_0$ | a | b | deviance | AIC |
|---|---|---|---|---|---|
| $CP_1$ | | -0.1059 ± 0.00044 | 0.1858 ± 0.00129 | 64390 | 82702 |
| $CP_2$ | | -1.1649 ± 0.00298 | 0.4897 ± 0.00346 | 55354 | 73667 |
| $CP_3$ | | -0.3651 ± 0.00117 | 0.7369 ± 0.00687 | 51706 | 70018 |
| $CP_4$ | 0.6426 ± 0.00662 | 0.08345 ± 0.00238 | -0.3886 ± 0.0014 | 50479 | 68793 |
| $CP_5$ | 0.8867 ± 0.01012 | 0.2411 0.00377 | -0.5604 ± 0.00298 | 45559 | 63874 |

$a$: mean ± standard error.

**Table 3. Deviance and AIC for each model in the calibration and validation sets.**

| Model | Cross-Pollination Model | Calibration Set | | Validation Set | |
|---|---|---|---|---|---|
| | | Deviance | AIC | Deviance | AIC |
| $CP_1$ | $CP^a = 0.1957 \times e^{[-0.1092(distance)]}$ | 43638 | 55686 | 22400 | 28679 |
| $CP_2$ | $CP = 0.5 \times distance^{-1.1782}$ | 36830 | 48879 | 19224 | 25503 |
| $CP_3$ | $CP = 0.7959 \times 10^{-0.3746\sqrt{distance}}$ | 34348 | 46396 | 18008 | 24286 |
| $CP_4$ | $CP = 0.7018 \times 10^{[0.0771\sqrt{FB}+(-0.3964)\sqrt{distance}]}$ | 33670 | 45720 | 17496 | 23777 |
| $CP_5$ | $CP = 0.003 + 0.9759 \times 10^{[(0.2460)\sqrt{FB}+(-0.5788)\sqrt{distance}]}$ | 29994 | 42044 | 15878 | 22158 |

*a*: cross-pollination (CP) rate (%).

and $CP_3$ as well as of *b* for $CP_4$ and $CP_5$ were negative, implying that CP rate (%) decreased as distance from the pollen source increased. However, the estimated values of *a* for $CP_4$ and $CP_5$ were positive, indicating that the FB may have enhanced pollen exchange. Moreover, in $CP_5$ and $CP_4$, the lower deviance and AIC compared with the other models indicated a closer fit for the complete dataset. However, to evaluate the model fitting results and compare the predictive abilities of the investigated models, the experimental data were randomly partitioned into a calibration set and validation set.

Table 3 presents the fitting ability of each CP model based on the deviance and AIC for calibration and validation data. When the fit was assessed using the calibration set, four of the models had deviance > 30000, and only one had deviance < 30000. The lowest deviance value was 29994 ($CP_5$). Moreover, four models had AIC > 45000, and one had AIC < 45000. The lowest AIC value was 42044 ($CP_5$).

For the validation set, the fitting abilities were similar to the fits evaluated using the calibration set. Model $CP_5$ exhibited optimal fitting ability with the lowest deviance value of 15878 and AIC value of 22158 and was followed by $CP_4$ (deviance = 17496, AIC = 23777; Table 3). The scatter plots of actual and predicted values in Fig 5 indicate the predictive abilities of the models based on the validation set. Only one model had r ≤ 0.7 ($CP_1$), four models had r > 0.7. The highest r value was 0.79 ($CP_5$), indicating that $CP_5$ had higher prediction abilities than the other models.

Accordingly, $CP_1$, $CP_2$, and $CP_3$ had worse fits and predictive abilities than the other models. $CP_5$ exhibited the optimal fit and predictive ability, followed by $CP_4$. Because the FB effect occurred in some fields, $CP_1$, $CP_2$, and $CP_3$ had worse fits and prediction abilities relative to the models that included the FB effect.

## Simulation results

Bootstrap simulations were implemented to compare the stabilities of the fits and predictive abilities of the CP models. The simulation method was repeated for 1,000 runs, where 1,000 bootstrap samples were generated during 1,000 runs. The means and SDs of the parameters were calculated for each CP model based on the bootstrap simulations (Table 4). In addition, the mean and SD of the deviance and AIC of the simulated calibration and validation sets were also calculated (Table 5). For the validation sets, the correlation coefficient was calculated to evaluate the model predictive ability.

The estimated values of parameters (Table 4) were similar to the results obtained from the complete observed data listed in Table 2. This indicated that the parameters estimated from the simulation data were identical to those estimated from the whole original data.

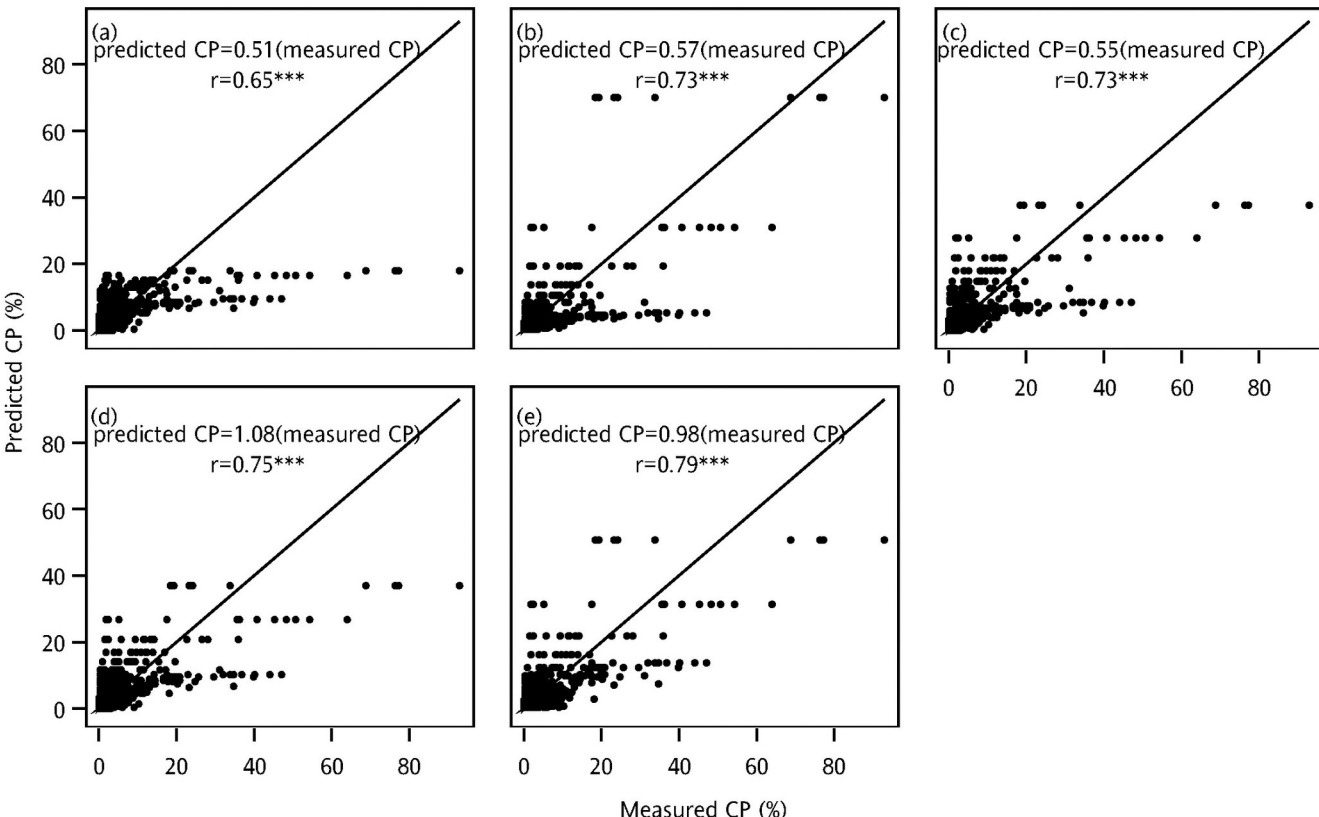

**Fig 5. Scatter plots of measured and predicted CP values in the validation set.** Models (a) $CP_1$, (b) $CP_2$, (c) $CP_3$, (d) $CP_4$, and (e) $CP_5$ (***: $p < .0001$).

The average deviance values of $CP_5$ and $CP_4$ were lower than those of the other models, indicating that the fits of $CP_5$ and $CP_4$ were closer to the data than were those of the other models. Among the other models, $CP_1$, $CP_2$, and $CP_3$ performed similarly in terms of average deviance. Furthermore, the SDs of deviance were higher in $CP_1$, $CP_2$, and $CP_3$ than in $CP_5$ and $CP_4$. This indicated that $CP_1$, $CP_2$, and $CP_3$ had worse fit and greater variability than $CP_5$ and $CP_4$ did. Moreover, the average AIC values were similar to the results for deviance (Table 5). $CP_5$ and $CP_4$ exhibited the lowest average AIC values, and they had smaller SDs compared with the other models. In summary, $CP_5$ and $CP_4$ had the optimal fit, and these fits were more stable than those of the other models.

**Table 4. Regression parameters for each model based on the simulation data.**

| Model | $P_0$ | a | b |
|---|---|---|---|
| $CP_1$ | | -0.1058 ± 0.0029 | 0.1856 ± 0.0112 |
| $CP_2$ | | -1.1653 ± 0.0161 | 0.4907 ± 0.0238 |
| $CP_3$ | | -0.3648 ± 0.0064 | 0.7359 ± 0.0473 |
| $CP_4$ | 0.6414 ± 0.0437[a] | 0.0836 ± 0.0127 | -0.3883 ± 0.0076 |
| $CP_5$ | 0.8861 ± 0.0624 | 0.2417 ± 0.0227 | -0.5607 ± 0.0172 |

*a*: mean ± standard deviation.

**Table 5. Deviance, AIC and r for each model based on the simulation data.**

| Model | Calibration Set | | Validation Set | | |
| --- | --- | --- | --- | --- | --- |
| | Deviance | AIC | Deviance | AIC | r |
| $CP_1$ | $42852 \pm 2029$[a] | $55064 \pm 2073$ | $21441 \pm 1387$ | $27546 \pm 1417$ | $0.651 \pm 0.017$ |
| $CP_2$ | $36887 \pm 1950$ | $49099 \pm 1992$ | $18452 \pm 1389$ | $24557 \pm 1419$ | $0.736 \pm 0.045$ |
| $CP_3$ | $34433 \pm 1486$ | $46645 \pm 1533$ | $17238 \pm 1033$ | $23343 \pm 1066$ | $0.739 \pm 0.027$ |
| $CP_4$ | $33576 \pm 1284$ | $45790 \pm 1333$ | $16845 \pm 904$ | $22953 \pm 939$ | $0.759 \pm 0.024$ |
| $CP_5$ | $30288 \pm 1060$ | $42502 \pm 1111$ | $15213 \pm 767$ | $21321 \pm 802$ | $0.796 \pm 0.027$ |

*a*: mean ± standard deviation.

The validation results were similar to those obtained from the calibration set (Table 5). This indicated that the fit and validation results were consistent and no overfitting occurred. Moreover, $CP_5$ and $CP_4$ remained more fitted than the other models. The SDs of deviance and AIC in $CP_5$ and $CP_4$ were also smaller than those in the other models. The $CP_5$ also presented the highest r (r = 0.796) among the models. The value of r indicated that the $CP_4$ and $CP_5$ performed the best predictive ability.

For the calibration and validation sets, the results of $CP_5$ and $CP_4$ were superior to those of the other models. Therefore, $CP_5$ and $CP_4$, which included FBs, had the closest fits, optimal predictive abilities, and most stable performance.

Additionally, the 95% PB confidence interval lengths (PBLs) of the average deviance and AIC values were used to further assess the stabilities of model fitting (Figs 6 and 7).

As indicated in Fig 6, the fits of $CP_5$ and $CP_4$ were more stable than those of the other models, resulting in smaller PBLs ($PBL_{CP5} = 4075$, $PBL_{CP4} = 5140$). Because the PB confidence intervals were overlapping, the $CP_5$ and $CP_4$ models exhibited similar performance. Most of

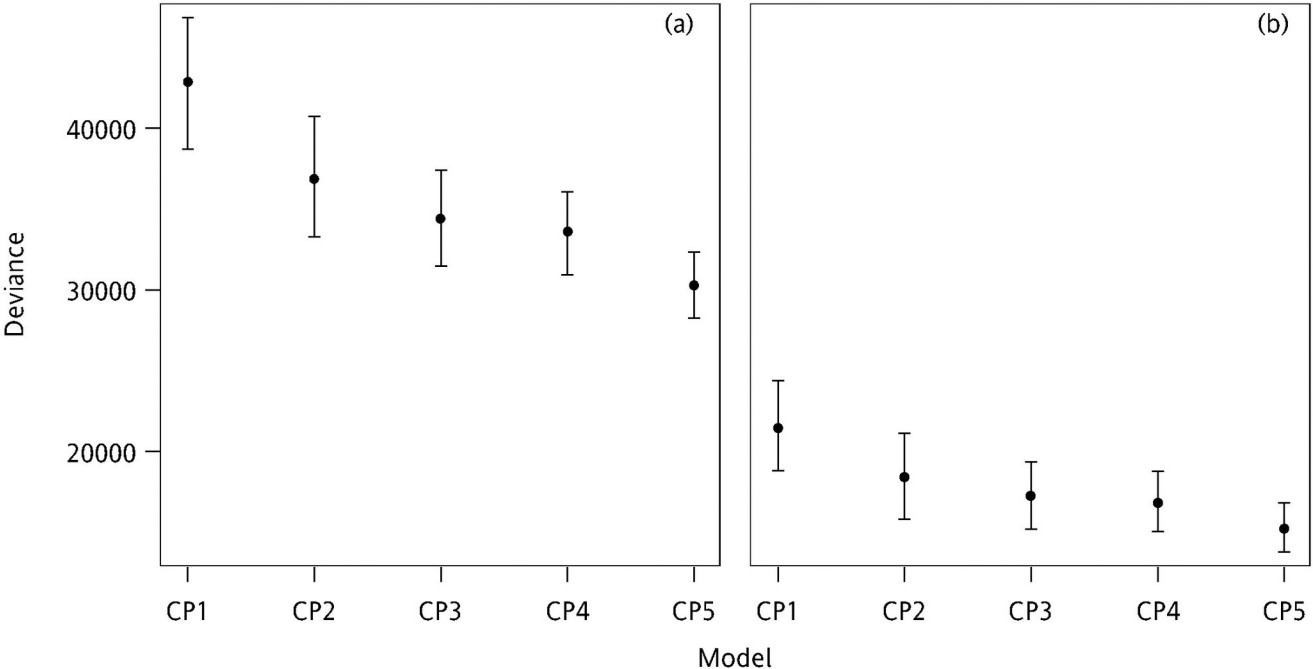

**Fig 6.** 95% PBLs of deviance based on the simulated (a) calibration and (b) validation sets for each model.

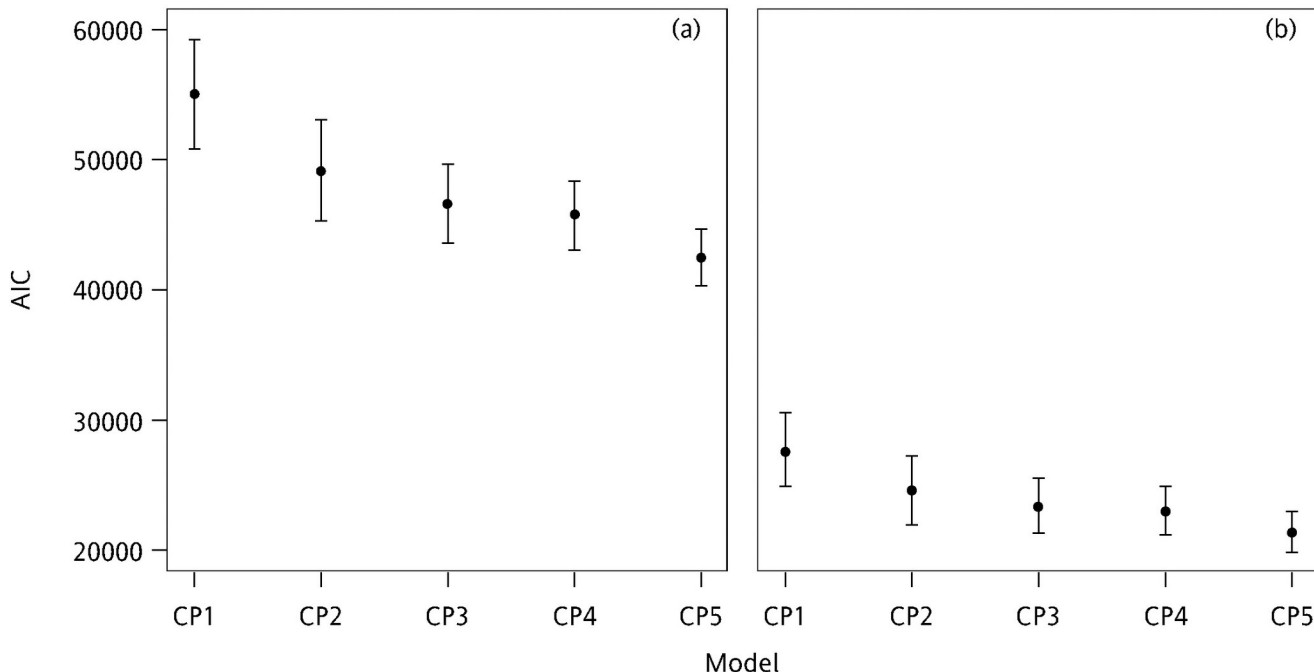

**Fig 7.** 95% PBLs of AIC based on the simulated (a) calibration and (b) validation sets for each model.

the models exhibited relatively larger PBLs except $CP_4$ and $CP_5$. Additionally, the results obtained from the validation set were similar to those obtained from the calibration set.

Moreover, the 95% PBLs of the average AIC values were similar to the results for deviance. $CP_5$ and $CP_4$ exhibited similar fits and were more stable than the other models (Fig 7). In addition, the results obtained from the validation and calibration sets were similar.

In summary, $CP_5$ and $CP_4$ exhibited the optimal stability, model fits, and predictive abilities. Moreover, the values of the regression parameters estimated based on the simulation data were similar to the values calculated using the complete observed data. Furthermore, the comparison results of deviance and AIC estimated using the simulation data were similar to those obtained from the complete observed data. Additionally, the results in the calibration and validation sets were similar.

## Conclusion

In Taiwan and most Asian countries, crop fields are often separated by roadways. Therefore, this study established a CP model for maize to describe the relationship between CP rate (%) and distance from a pollen source; this model was suitable at the field-scale level in southern Taiwan. Three models commonly applied in gene flow studies and empirical modeling approaches including the FB effect were investigated. The results indicated that the models that omitted the FB yielded poor performance. By contrast, models that included FB had improved fits and predictive abilities as well as the greatest stability.

Previous studies have examined the models used in the present study, but their results differ from those of the present study because of differences in experimental designs. Ma et al. [4] used the $CP_2$ model to describe downwind and upwind data. Their pollen source (0.07 ha) was placed in the center of the field and surrounded by the pollen recipient (0.68–1 ha). Their $R^2$ values for the fitting results were 0.64 for downwind areas and 0.58 for upwind areas.

Therefore, Ma et al. [4] concluded that the $CP_2$ model was suitable for describing the relationship between CP rate (%) and distance from the pollen source. In addition, Della Porta et al. [5] used the $CP_1$, $CP_2$, and $CP_3$ models for various field designs, barriers, positions, and distances from the pollen source to perform model fitting. Their results revealed that the $CP_2$ model provided the optimal fit ($R^2 = 0.89$–$0.99$) when the recipient neighbored the pollen source without a barrier. In general, pollen flow tendencies were similar for a fixed FB size. Therefore, the $CP_2$ model exhibited similarly high performance when the FB size was fixed; resulting in $R^2$ values extremely close to 1.

One of the aims of this study was to investigate the effects of FBs on the gene flow tendencies between the adjacent fields. According to the estimates of the regression parameters in the empirical and simulation analyses as well as the results of the gene flow trends, the CP rate (%) decreased as the distance from the pollen source increased. The presence of an FB enhanced pollen exchange in the row of the pollen recipient closest to the pollen source and facilitated long-distance pollination events [18]. However, when the distance from the pollen source increased, the average CP rate (%) in the presence of an FB may have been lower than that in the absence of an FB (Table 1).

Considering isolation buffers (e.g., FBs) together, the $CP_5$ model exhibited the optimal stability, fit, and predictive ability. The $CP_4$ model also yielded performance similar to that of $CP_5$. However, the AP must be set to 0% for the agricultural environment in Taiwan. Therefore, although the $CP_5$ model exhibited the optimal performance, after setting AP to 0%, $CP_4$ appropriately modeled the data obtained from southern Taiwan and was recommended in this study.

In addition to exploring the relationship between CP and the distances from the pollen source, studies on the CP models for GM maize have recently included other influential factors to improve the completeness and accuracy of predictive models. Loos et al. [19] used a Gaussian plume model to simulate the pollen transport of maize in and from plant canopies. This semi-empirical approach combined the atmospheric diffusion equation with the Lagrangian method. To describe the trends of maize pollen dispersal, Klein et al. [20] used individual dispersal functions containing biological parameters (i.e., difference in height between male and female flowers) and aerodynamic parameters (i.e., pollen settling velocity, wind speed, and air turbulence). Goggi et al. [21] combined an exponential model with a linear equation to establish a maize CP model. The exponential model described the relationship between the CP rate (%) and the distance from the pollen source, whereas the linear equation described the relationship between wind speed, wind direction, and distance.

In addition to the distance from the pollen source and FB considered in this study, meteorological and biological factors have been gradually introduced in CP models. Therefore, these factors should be included in future studies to further improve the fit and predictive ability of the proposed models for investigating gene flow and CP. In this study, only three experiments data were used to estimate the parameters of models. For the further study, more replications of experiment should be needed to establish a more robust model.

## Supporting information

**S1 Table. Regression parameters, deviance, and AIC for the complete dataset in models with parameter $P_0$.**
(DOCX)

**S2 Table. Deviance and AIC for models with parameter $P_0$ in the calibration and validation sets.**
(DOCX)

**S3 Table. Deviance, AIC and r for models with parameter $P_0$ based on the simulation data.**
(DOCX)

**S1 File.**
(DOCX)

**S1 Fig.**
(TIFF)

## Acknowledgments

This manuscript was edited by Wallace Academic Editing.

## Author Contributions

**Conceptualization:** Bo-Jein Kuo, Wen-Shin Lin.

**Data curation:** Yun-Syuan Jhong, Tien-Joung Yiu.

**Formal analysis:** Yun-Syuan Jhong.

**Funding acquisition:** Bo-Jein Kuo.

**Investigation:** Yun-Syuan Jhong, Tien-Joung Yiu, Yuan-Chih Su.

**Methodology:** Yun-Syuan Jhong, Tien-Joung Yiu, Yuan-Chih Su.

**Project administration:** Bo-Jein Kuo, Wen-Shin Lin.

**Resources:** Yun-Syuan Jhong, Tien-Joung Yiu.

**Software:** Yun-Syuan Jhong, Yuan-Chih Su.

**Supervision:** Bo-Jein Kuo, Wen-Shin Lin.

**Validation:** Yun-Syuan Jhong, Yuan-Chih Su.

**Visualization:** Yun-Syuan Jhong, Yuan-Chih Su.

**Writing – original draft:** Bo-Jein Kuo, Yun-Syuan Jhong, Yuan-Chih Su, Wen-Shin Lin.

**Writing – review & editing:** Bo-Jein Kuo, Yun-Syuan Jhong, Yuan-Chih Su, Wen-Shin Lin.

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
