## [Decision Letter · Decision Letter 0]

21 Sep 2020

PONE-D-20-23403

Bootstrap simulations for evaluating the model estimation of the extent of cross-pollination in maize at the field-scale level

PLOS ONE

Dear Dr. Lin,

Thank you for submitting your manuscript to PLOS ONE. After careful consideration, we feel that it has merit but does not fully meet PLOS ONE’s publication criteria as it currently stands. Therefore, we invite you to submit a revised version of the manuscript that addresses the points raised during the review process.

We look forward to receiving your revised manuscript.

Kind regards,

Mehdi Rahimi, Ph.D.

Academic Editor

PLOS ONE

Journal Requirements:

2. Please upload a new copy of Figure 5 as the detail is not clear. Please follow the link for more information: https://blogs.plos.org/plos/2019/06/looking-good-tips-for-creating-your-plos-figures-graphics/" https://blogs.plos.org/plos/2019/06/looking-good-tips-for-creating-your-plos-figures-graphics/

4.Thank you for stating the following in the Acknowledgments Section of your manuscript:

[This research is partially supported by the Ministry of Science and Technology under

Grant Number 108-2634-F-005-003 through “Pervasive AI Research (PAIR) Labs,

Taiwan”, and “Innovation and Development Center of Sustainable Agriculture” from

The Featured Areas Research Center Program within the framework of the Higher

Education Sprout Project by the Ministry of Education (MOE) in Taiwan.].

The authors received no specific funding for this work.

We note that one or more of the authors is affiliated with the funding organization (“Pervasive AI Research (PAIR) Labs,

Taiwan), indicating the funder may have had some role in the design, data collection, analysis or preparation of your manuscript for publication; in other words, the funder played an indirect role through the participation of the co-authors.

If the funding organization did not play a role in the study design, data collection and analysis, decision to publish, or preparation of the manuscript and only provided financial support in the form of authors' salaries and/or research materials, please review your statements relating to the author contributions, and ensure you have specifically and accurately indicated the role(s) that these authors had in your study in the Author Contributions section of the online submission form. Please make any necessary amendments directly within this section of the online submission form.  Please also update your Funding Statement to include the following statement: “The funder provided support in the form of salaries for authors [insert relevant initials], but did not have any additional role in the study design, data collection and analysis, decision to publish, or preparation of the manuscript. The specific roles of these authors are articulated in the ‘author contributions’ section.”

If the funding organization did have an additional role, please state and explain that role within your Funding Statement.

Please also provide an updated Competing Interests Statement declaring this commercial affiliation along with any other relevant declarations relating to employment, consultancy, patents, products in development, or marketed products, etc.  

Reviewers' comments:

Reviewer's Responses to Questions

**Comments to the Author**

1. Is the manuscript technically sound, and do the data support the conclusions?

Reviewer #1: Yes

Reviewer #2: Yes

2. Has the statistical analysis been performed appropriately and rigorously? 

Reviewer #1: I Don't Know

Reviewer #2: Yes

3. Have the authors made all data underlying the findings in their manuscript fully available?

Reviewer #1: No

Reviewer #2: Yes

4. Is the manuscript presented in an intelligible fashion and written in standard English?

Reviewer #1: Yes

Reviewer #2: Yes

5. Review Comments to the Author

Reviewer #1: This paper studies the dispersal of maize pollen from a pollen source as a function of distance. Three fields are used for study. Seven nonlinear regression models are fitted. A bootstrap analysis is performed to provide confidence limits. Furthermore, cross validation is conducted.

The statistical methodology needs to be more accurately described. Some detailed comments follow.

(1) The study obviously involves a large number of plots per field. But it remains unclear what the dimension of the plots are and how they are distributed across the field. A map showing the field layout would be very useful.

(2) It is not clear from M&M whether the regression models were fitted per field or across fields.

(3) Missing plots were estimated using some clustering method, but it is unclear just why missing values had to be replaced (lines 294-297). Only a very small number or plots was missing, and then nonlinear models look quite simple, so I do no see any need to replace missing values.

(4) I read lines 260-263 several times, but could not understand the meaning. Can this be reworded? How can the gene flow be reduced by fitting a model? I do not understand.

(5) For the bootstrapping, it is unclear if this was done by field or for all data across lumped together across the three fields.

(6) How was the nonlinear regression done, what procedure in SAS was used? Was regression done by field or across fields? In the latter case, how was heterogeneity between fields taken into account?

(7) How was the cross validation done? Which SAS procedure? How was the data split into calibration and validation sets? It would be most meaning full, e.g., to use two fields for calibration and one for validation.

(8) Strictly speaking the number of true replications is given by the number of fields, whereas the many plots per field constitute pseudo-replications. This limitation of thee study deserves to be mentioned and critically discussed.

(9) I can't read the figures, there is too much blemish.

Reviewer #2: The authors of this manuscripts present bootstrap simulations for evaluating the model estimation of the extent of cross-pollination in maize at the field-scale level. It is an interesting manuscript that presents an important case study of the co-existence between genetically modified and non-genetically modified systems in an open environment. One struggle i faced while going through the manuscript was the need of more effort in clarity in writing and correcting grammatical mistakes, and succinctly providing enough details of their findings in their manuscript, that is, there were several incidences, where the information were repetitive with no additional information, also excessive use of abbreviation which makes it hard to remember all their full forms and hindering in the flow of the read. Overall, manuscript has an interesting case study and meets this journal standards for publication.

I have listed my specific comments with line numbers below:

L 51: Please provide the citation

L 54: 'us' should be 'is'?

L 56-57: Its a confusing sentence. Should it be interest or concern?

L 77: The word 'pollution' is not here in this sentence. May be use contamination instead?

L 133: "Polluted" is not the correct use here

L 147-149: Can cross-incompatibility trait assist in resolving this issue? For example check this paper -- Evans, M., Kermicle, J. Teosinte crossing barrier1, a locus governing hybridization of teosinte with maize. Theor Appl Genet 103, 259–265 (2001). https://doi.org/10.1007/s001220100549

L 193-194: the definition about the xenia effect does not make sense. Please correct.

L 200: Please provide experimental design of the experiments

L 217: Out of curiosity, Were there any pollinating insects such as bees that could potentially cross pollinate between two

varieties of corn?

L 220-221: Why only this duration of time was chosen?

L 233-234: How many corn ears per plant were counted?

L 241-243: Corn plants at the beginning and at the end of each row have a higher probability of containing CP kernels as they are more exposed than the plants in the middle of the row. I hope this scenario was considered while sampling, since it can biased the analysis and the results. similarly, its always advised to harvest top ear of the corn plant for the same reason.

L 260: Please correct the spelling of trial

L 317: In statistical analysis, what were the fixed/random effects in the model?

L 318-319: Will the SAS code be published or deposited in any publicly accessible repositories?

6. PLOS authors have the option to publish the peer review history of their article (what does this mean?). If published, this will include your full peer review and any attached files.

Reviewer #1: No

Reviewer #2: No

---

## [Author Response · Author response to Decision Letter 0]

19 Oct 2020

Response:

Thanks for the reminder from the editorial office. The manuscript has been checked follow the PLOS ONE's style requirements.

2. Please upload a new copy of Figure 5 as the detail is not clear. Please follow the link for more information: https://blogs.plos.org/plos/2019/06/looking-good-tips-for-creating-your-plos-figures-graphics/" https://blogs.plos.org/plos/2019/06/looking-good-tips-for-creating-your-plos-figures-graphics/

Response:

Thanks for the reminder from the editorial office. The new copy of figure 5 with higher resolution has been upload.

Response:

Thanks for the reminder from the editorial office. In our manuscript, the data referred by the phrase “data not shown” in LINE 392 and LINE420 are not a core part of the research being presented in our study, and in LINE 443, the value of r could be calculated by Table 3. Therefore, the phrase “data not shown” has been removed. 

4.Thank you for stating the following in the Acknowledgments Section of your manuscript:

[This research is partially supported by the Ministry of Science and Technology under Grant Number 108-2634-F-005-003 through “Pervasive AI Research (PAIR) Labs, Taiwan”, and “Innovation and Development Center of Sustainable Agriculture” from The Featured Areas Research Center Program within the framework of the Higher Education Sprout Project by the Ministry of Education (MOE) in Taiwan.].

The authors received no specific funding for this work.

We note that one or more of the authors is affiliated with the funding organization (“Pervasive AI Research (PAIR) Labs, Taiwan), indicating the funder may have had some role in the design, data collection, analysis or preparation of your manuscript for publication; in other words, the funder played an indirect role through the participation of the co-authors.

If the funding organization did not play a role in the study design, data collection and analysis, decision to publish, or preparation of the manuscript and only provided financial support in the form of authors' salaries and/or research materials, please review your statements relating to the author contributions, and ensure you have specifically and accurately indicated the role(s) that these authors had in your study in the Author Contributions section of the online submission form. Please make any necessary amendments directly within this section of the online submission form. Please also update your Funding Statement to include the following statement: “The funder provided support in the form of salaries for authors [insert relevant initials], but did not have any additional role in the study design, data collection and analysis, decision to publish, or preparation of the manuscript. The specific roles of these authors are articulated in the ‘author contributions’ section.”

If the funding organization did have an additional role, please state and explain that role within your Funding Statement.

Please also provide an updated Competing Interests Statement declaring this commercial affiliation along with any other relevant declarations relating to employment, consultancy, patents, products in development, or marketed products, etc. 

Response:

Thanks for the reminder from the editorial office. In cover letter, we would like to update our Funding Statement, because we will only get the financial support for the Article Processing Charge (APC) for our article published. In addition, the funding organization did not have an additional role in the study design, data collection and analysis, decision to publish, or preparation of the manuscript and did not provide financial support in the form of authors' salaries and/or research materials. The academic affiliation does not alter our adherence to PLOS ONE policies on sharing data and materials. 

Reviewers' comments:

Reviewer's Responses to Questions

Comments to the Author

1. Is the manuscript technically sound, and do the data support the conclusions?

Reviewer #1: Yes

Reviewer #2: Yes

2. Has the statistical analysis been performed appropriately and rigorously?

Reviewer #1: I Don't Know

Reviewer #2: Yes

3. Have the authors made all data underlying the findings in their manuscript fully available?

Reviewer #1: No

Reviewer #2: Yes

4. Is the manuscript presented in an intelligible fashion and written in standard English?

Reviewer #1: Yes

Reviewer #2: Yes

5. Review Comments to the Author

Reviewer #1: This paper studies the dispersal of maize pollen from a pollen source as a function of distance. Three fields are used for study. Seven nonlinear regression models are fitted. A bootstrap analysis is performed to provide confidence limits. Furthermore, cross validation is conducted.

The statistical methodology needs to be more accurately described. Some detailed comments follow.

(1) The study obviously involves a large number of plots per field. But it remains unclear what the dimension of the plots are and how they are distributed across the field. A map showing the field layout would be very useful.

Response to Reviewers: 

Thanks for the reviewer’s comment. We have added the example of the layout and the dimension of sampling plots in Fig.1.

(2) It is not clear from M&M whether the regression models were fitted per field or across fields.

Response to Reviewers:

Thanks for the reviewer’s comment. The PMFG model were fitted across fields. As shown in Table 2, the results of regression parameters, RMSE, and R2 were fitted the complete dataset. The statement was also corrected and inserted into LINE 426.

(3) Missing plots were estimated using some clustering method, but it is unclear just why missing values had to be replaced (lines 294-297). Only a very small number or plots was missing, and then nonlinear models look quite simple, so I do no see any need to replace missing values.

Response to Reviewers:

Thanks for the reviewer’s comment. In our study, the P0, the CP rate (%) of the first row of the pollen recipient field, is required using the PMFG models. However, the collected data will be lost resulted from the environmental conditions of the site. Therefore, to ensure that the PMFG models can be evaluated, the k-nearest neighbor algorithm (k-NN) proposed by Šuštar-Vozlič et al. 2010 [13] does need to be used in our study. 

(4) I read lines 260-263 several times, but could not understand the meaning. Can this be reworded? How can the gene flow be reduced by fitting a model? I do not understand.

Response to Reviewers:

Thanks for the reviewer’s comment. The statement means that as reported in Gustafson et al. 2006 [12], the trend of gene flow within recipient or border rows could be reduced as represented as the coefficient of the proportionality constant, and twice as effective as the ID (the unplanted isolation buffer). Then, in our study, the proportionality constants for FB and the distance within the recipient field were re-estimated based on the data from field experiments conducted. The statement was also corrected and inserted into LINES 260 to 263. 

(5) For the bootstrapping, it is unclear if this was done by field or for all data across lumped together across the three fields.

Response to Reviewers:

Thanks for the reviewer’s comment. For the bootstrapping, random sampling with replacement was applied to the actual data set across lumped together across the three fields. The statement was also corrected and inserted into LINE 309.

(6) How was the nonlinear regression done, what procedure in SAS was used? Was regression done by field or across fields? In the latter case, how was heterogeneity between fields taken into account?

Response to Reviewers:

Thanks for the reviewer’s comment. In our study, PROC NLIN was used for constructing the PMFG model across the three fields. On the other hand, due to the facts that the three fields were set in the same location, Puzih City, Chiayi County and the trend of gene flow were all similar, three data sets collected from the same field were combined and used to evaluate the performance of the PMFG model.

(7) How was the cross validation done? Which SAS procedure? How was the data split into calibration and validation sets? It would be most meaning full, e.g., to use two fields for calibration and one for validation.

Response to Reviewers:

Thanks for the reviewer’s comment. Three data sets collected from the same field were combined as the complete data. Then, the complete experimental data were randomly partitioned into two sets, a calibration set (two thirds of the samples) and a validation set (one third of the samples), for fitting to the model and validating their predictive abilities, respectively. 

(8) Strictly speaking the number of true replications is given by the number of fields, whereas the many plots per field constitute pseudo-replications. This limitation of the study deserves to be mentioned and critically discussed.

Response to Reviewers:

Thanks for the reviewer’s comment. We agree that the true replication is the number of fields. In our study, the main purpose is modeling the gene flow. Therefore, the experiment was designed for establishing the regression models. 

(9) I can't read the figures, there is too much blemish.

Response to Reviewers:

Thanks for the reviewer’s comment. We have modified the resolution of the figures.

Reviewer #2: The authors of this manuscripts present bootstrap simulations for evaluating the model estimation of the extent of cross-pollination in maize at the field-scale level. It is an interesting manuscript that presents an important case study of the co-existence between genetically modified and non-genetically modified systems in an open environment. One struggle i faced while going through the manuscript was the need of more effort in clarity in writing and correcting grammatical mistakes, and succinctly providing enough details of their findings in their manuscript, that is, there were several incidences, where the information were repetitive with no additional information, also excessive use of abbreviation which makes it hard to remember all their full forms and hindering in the flow of the read. Overall, manuscript has an interesting case study and meets this journal standards for publication.

I have listed my specific comments with line numbers below:

L 51: Please provide the citation

Response to Reviewers:

Thanks for the reviewer’s comment. The citation was the same as [1].

L 54: 'us' should be 'is'?

Response to Reviewers:

Thanks for the reviewer’s comment. The sentence has been corrected. 

L 56-57: Its a confusing sentence. Should it be interest or concern?

Response to Reviewers:

Thanks for the reviewer’s comment. The sentence has been corrected.

L 77: The word 'pollution' is not here in this sentence. May be use contamination instead?

Response to Reviewers:

Thanks for the reviewer’s comment. In the gene flow study, the word ‘pollution’ means gene exchange between different varieties through pollen.

L 133: "Polluted" is not the correct use here

Response to Reviewers:

Thanks for the reviewer’s comment. The sentence has been corrected as ‘reduce the risk of conventional crops being hybridized by GM crops’ in LINE 133.

L 147-149: Can cross-incompatibility trait assist in resolving this issue? For example check this paper -- Evans, M., Kermicle, J. Teosinte crossing barrier1, a locus governing hybridization of teosinte with maize. Theor Appl Genet 103, 259–265 (2001). https://doi.org/10.1007/s001220100549

Response to Reviewers:

Thanks for the reviewer’s comment. In our study, the performance of PMFG models is the objective evaluated by the actual field data, not using molecular biotechnology for isolating one category of commercial varieties from another.

L 193-194: the definition about the xenia effect does not make sense. Please correct.

Response to Reviewers:

Thanks for the reviewer’s comment. The sentence has been corrected as ‘The xenia effect of the maize is caused by the effect of the different pollen source gene resulted in endosperms on the development of the seeds.’, as shown in LINES 193 to 195.

L 200: Please provide experimental design of the experiments

Response to Reviewers:

Thanks for the reviewer’s comment. The detail field design were shown in Fig.1 and noted in LINE 200.

L 217: Out of curiosity, Were there any pollinating insects such as bees that could potentially cross pollinate between two varieties of corn?

Response to Reviewers:

Thanks for the reviewer’s comment. Maize is anemophilous flower crop. Therefore, the probability of cross pollinate between two varieties of corn by the pollinating insects is much less than by the wind.

L 220-221: Why only this duration of time was chosen?

Response to Reviewers:

Thanks for the reviewer’s comment. In Taiwan and most Asian countries, crop fields are often separated by intervening roadways. Moreover, in Taiwan, there are two suitable seasons for planting maize begin in May and October, respectively. Therefore, the study decided in these two duration of time. 

L 233-234: How many corn ears per plant were counted?

Response to Reviewers:

Thanks for the reviewer’s comment. In Taiwan, there might be one to three corn ears per plant. However, the first ear has commodity value. Therefore, the first ear was investigated in our study.

L 241-243: Corn plants at the beginning and at the end of each row have a higher probability of containing CP kernels as they are more exposed than the plants in the middle of the row. I hope this scenario was considered while sampling, since it can biased the analysis and the results. similarly, its always advised to harvest top ear of the corn plant for the same reason.

Response to Reviewers:

Thanks for the reviewer’s comment. Indeed, Corn plants at the beginning of each row have a higher probability of containing CP kernels than the plants in the middle or the end of the row. Therefore, when designing the sampling plant, the sampling frequency changed with the distance from the pollen source.

L 260: Please correct the spelling of trial

Response to Reviewers:

Thanks for the reviewer’s comment. The word has been corrected in LINE 262.

L 317: In statistical analysis, what were the fixed/random effects in the model?

Response to Reviewers:

Thanks for the reviewer’s comment. In the PMFG model, the effect of FB and distance were evaluated, as the random effects in the model.

L 318-319: Will the SAS code be published or deposited in any publicly accessible repositories?

Response to Reviewers:

Thanks for the reviewer’s comment. In the future, the SAS code in our study will be provided for researchers in need.

---

## [Decision Letter · Decision Letter 1]

16 Nov 2020

PONE-D-20-23403R1

Bootstrap simulations for evaluating the model estimation of the extent of cross-pollination in maize at the field-scale level

PLOS ONE

Dear Dr. Lin,

Thank you for submitting your manuscript to PLOS ONE. After careful consideration, we feel that it has merit but does not fully meet PLOS ONE’s publication criteria as it currently stands. Therefore, we invite you to submit a revised version of the manuscript that addresses the points raised during the review process.

We look forward to receiving your revised manuscript.

Kind regards,

Mehdi Rahimi, Ph.D.

Academic Editor

PLOS ONE

Reviewers' comments:

Reviewer's Responses to Questions

**Comments to the Author**

1. If the authors have adequately addressed your comments raised in a previous round of review and you feel that this manuscript is now acceptable for publication, you may indicate that here to bypass the “Comments to the Author” section, enter your conflict of interest statement in the “Confidential to Editor” section, and submit your "Accept" recommendation.

Reviewer #1: (No Response)

Reviewer #2: All comments have been addressed

2. Is the manuscript technically sound, and do the data support the conclusions?

Reviewer #1: (No Response)

Reviewer #2: Yes

3. Has the statistical analysis been performed appropriately and rigorously? 

Reviewer #1: (No Response)

Reviewer #2: Yes

4. Have the authors made all data underlying the findings in their manuscript fully available?

Reviewer #1: (No Response)

Reviewer #2: Yes

5. Is the manuscript presented in an intelligible fashion and written in standard English?

Reviewer #1: (No Response)

Reviewer #2: Yes

6. Review Comments to the Author

Reviewer #1: L239: CP% is based on counts and as such is bound to display heterogeneity of variance. Residuals need to be carefully checked if simple nonlinear least squares is used and graphical evidence of a good fit based on residual plots should be provided. Alternatively, if residuals indicate problems, the nonlinear models can be fit using a generalized nonlinear regression package allowing a Poisson or Binomial error distribution to be assumed.

L296: I do not think the model requires observed data on P0: P0 is just another parameter that actually needs to be estimated. Thus, I also do not see a need to impute missing data at the first row of the pollen field.

L337: What are a and b here? Parameters of the nonlinear models, or are you fitting a regression of observed versus fitted data here and a and b are intercept and slope, respectively?

I had suggested that field is the experimental unit, so the whole study only has three true replications. the authors acknowledge this in their response, but their main text does not comment on this limitation. I think it would be important to do so.

Reviewer #2: Authors have addressed all my questions very well. However, the figures in the revised version still had a poor resolution such as axis labels and especially equations on figure 6, please fix them.

7. PLOS authors have the option to publish the peer review history of their article (what does this mean?). If published, this will include your full peer review and any attached files.

Reviewer #1: No

Reviewer #2: No

---

## [Author Response · Author response to Decision Letter 1]

13 Dec 2020

Reviewer #1: L239: CP% is based on counts and as such is bound to display heterogeneity of variance. Residuals need to be carefully checked if simple nonlinear least squares is used and graphical evidence of a good fit based on residual plots should be provided. Alternatively, if residuals indicate problems, the nonlinear models can be fit using a generalized nonlinear regression package allowing a Poisson or Binomial error distribution to be assumed.

Response: Thanks for the reviewer’s comment. We followed your suggestion and checked the residuals, carefully. It was found that the graphical evaluation of goodness of fit indicated that variance was heterogeneous (Fig. 1a-g). 

The fitting result of models with Poisson error distribution indicated a partial improvement of the assumption of homoscedastic (Fig 2a-e). Although the characteristic of pollen dispersal data still makes some dispersions, for the further analysis, the fitting of nonlinear models with Poisson error distribution were used to replace the normal error distribution in our previous manuscript.

More detail interpretation about Figs. 1a-g and Figs. 2a-e were shown in the file of "response to reviewer".

L296: I do not think the model requires observed data on P0: P0 is just another parameter that actually needs to be estimated. Thus, I also do not see a need to impute missing data at the first row of the pollen field.

Response: Thanks for the reviewer’s comment. Because the CP rate at the edge of the pollen recipient field vary with different cultivation scenarios, P0 should be calculated separately from different experiments in this study. The model may have a poor performance, if P0 is estimated using the data combining three experiments. Based on the data collected on-site in Taiwan, a pollen dispersion model suitable for use in Taiwan is constructed. In the future, under the condition that P0 can be calculated, CP% under a certain distance can be evaluated.

L337: What are a and b here? Parameters of the nonlinear models, or are you fitting a regression of observed versus fitted data here and a and b are intercept and slope, respectively?

Response: Thanks for the reviewer’s comment. The a and b are the parameters of the nonlinear models.

I had suggested that field is the experimental unit, so the whole study only has three true replications. the authors acknowledge this in their response, but their main text does not comment on this limitation. I think it would be important to do so.

Response: Thanks for the reviewer’s comment. We have described this limitation at the end of Conclusion (lines 602-605).

Reviewer #2: Authors have addressed all my questions very well. However, the figures in the revised version still had a poor resolution such as axis labels and especially equations on figure 6, please fix them.

Response: Thanks for the reviewer’s comment. We have fixed some figures that axis labels and equations are unclear. In addition, figures were assessed by PACE tool to check the technical requirements. The high-resolution figure file can be download by clicking the link at the upper right of figure page.

---

## [Decision Letter · Decision Letter 2]

21 Dec 2020

PONE-D-20-23403R2

Bootstrap simulations for evaluating the model estimation of the extent of cross-pollination in maize at the field-scale level

PLOS ONE

Dear Dr. Lin,

Thank you for submitting your manuscript to PLOS ONE. After careful consideration, we feel that it has merit but does not fully meet PLOS ONE’s publication criteria as it currently stands. Therefore, we invite you to submit a revised version of the manuscript that addresses the points raised during the review process.

We look forward to receiving your revised manuscript.

Kind regards,

Mehdi Rahimi, Ph.D.

Academic Editor

PLOS ONE

Reviewers' comments:

Reviewer's Responses to Questions

**Comments to the Author**

1. If the authors have adequately addressed your comments raised in a previous round of review and you feel that this manuscript is now acceptable for publication, you may indicate that here to bypass the “Comments to the Author” section, enter your conflict of interest statement in the “Confidential to Editor” section, and submit your "Accept" recommendation.

Reviewer #1: (No Response)

2. Is the manuscript technically sound, and do the data support the conclusions?

Reviewer #1: (No Response)

3. Has the statistical analysis been performed appropriately and rigorously? 

Reviewer #1: No

4. Have the authors made all data underlying the findings in their manuscript fully available?

Reviewer #1: (No Response)

5. Is the manuscript presented in an intelligible fashion and written in standard English?

Reviewer #1: Yes

6. Review Comments to the Author

Reviewer #1: The authors insist that imputation of P0 is nececcary prior to fitting some of their models. I can only re-iterate that I think P0 is a parameter that should be estimated simultaneously with the others in order to obtain a valid inference. The authors could compare the estimates of P0 resulting from this with their imputed value of P0. Also note that P0 in the these models has about the same status as b in some of their simpler models.

7. PLOS authors have the option to publish the peer review history of their article (what does this mean?). If published, this will include your full peer review and any attached files.

Reviewer #1: No

---

## [Author Response · Author response to Decision Letter 2]

23 Dec 2020

Response: Thanks for the reviewer’s comment and remind. Each model used in our study was the empirical model cited from the previous studies. And we want to evaluate the optimal mode suitable for use in Taiwan. As remind by reviewer, we also fitted CP4 by taking P0 as a parameter (see Table below). However, the fitting performance of CP4 with estimated P0 was worse than the fitting performance of CP4 with imputed P0. Therefore, according to the fitting performance we tend to keep the original method in our study. Moreover, as pointed by reviewer, the estimation of b in CP3 was very similar the value of P0 without FB in 2009-2, as shown in Tables 2, 3, and 4 in manuscript. On the other hand, we also used CP4 model by taking P0 into consideration. As shown by the simulation results, the SD of deviance were higher in CP3 than in CP4 using P0. This indicated that CP3 had worse fit and greater variability than CP4 did. Moreover, the average AIC values were similar to the results for deviance in Table 5. As descripted in the manuscript, CP7 and CP4 exhibited the lowest average AIC values, and they had smaller SDs compared with the other models. According to the conclusion, based on the data collected on-site in Taiwan, a pollen dispersion model suitable for use in Taiwan is constructed. In the future, under the condition that P0 can be calculated, CP% under a certain distance can be evaluated and provided for decision making in policy.

Table Comparison of parameters and fitting performance of CP4 with imputed P0 and estimated P0.

Model P0 a b deviance AIC

CP4 0.2122 -0.3905 41651 59951

CP4_P^a 0.6423 0.0835 -0.3886 50325 68626

a: P0 was estimated as a model parameter

---

## [Decision Letter · Decision Letter 3]

31 Dec 2020

PONE-D-20-23403R3

Bootstrap simulations for evaluating the model estimation of the extent of cross-pollination in maize at the field-scale level

PLOS ONE

Dear Dr. Lin,

Thank you for submitting your manuscript to PLOS ONE. After careful consideration, we feel that it has merit but does not fully meet PLOS ONE’s publication criteria as it currently stands. Therefore, we invite you to submit a revised version of the manuscript that addresses the points raised during the review process.

We look forward to receiving your revised manuscript.

Kind regards,

Mehdi Rahimi, Ph.D.

Academic Editor

PLOS ONE

Reviewers' comments:

Reviewer's Responses to Questions

**Comments to the Author**

1. If the authors have adequately addressed your comments raised in a previous round of review and you feel that this manuscript is now acceptable for publication, you may indicate that here to bypass the “Comments to the Author” section, enter your conflict of interest statement in the “Confidential to Editor” section, and submit your "Accept" recommendation.

Reviewer #1: (No Response)

2. Is the manuscript technically sound, and do the data support the conclusions?

Reviewer #1: (No Response)

3. Has the statistical analysis been performed appropriately and rigorously? 

Reviewer #1: (No Response)

4. Have the authors made all data underlying the findings in their manuscript fully available?

Reviewer #1: (No Response)

5. Is the manuscript presented in an intelligible fashion and written in standard English?

Reviewer #1: (No Response)

6. Review Comments to the Author

Reviewer #1: The authors assert that the deviance is better when P0 is imputed than when it is estimated. This suggests that something is wrong. The authors are using maximum likelihood estimation, so the value for P0 they find by that method should maximize the likelihood, and hence minimize the deviance. This can't possibly be beat by imputing P0 in any way. All of this is assuming, of course, that both times they are using the exact same data and the exact same likelihood function, which is necessary if you want to use deviance or AIC to compare models.

The authors state that they fitted the models as generalized linear models (GLM) but the models as stated in equations (2) to (8) are not generalized linear models. Several of them can be turned into a GLM upon taking the logarithm, and I am assuming this is what the authors have done. This would need to be explained succinctly. The parameter P0 of the model CP4 will be an intercept, but on the log-scale. Note that models CP6 and CP7 are not GLMs so it is not clear what was done to fit these. Since the authors are using SAS, they could use the NLMIXED procedure to fit these model (and other models for comparison to make sure they get the likelihood right). At any rate, it would be good to tell readers how exactly the different models were fitted in SAS.

One further issue: The authors assert that they assessed "predictive ability" by deviance and AIC. These are not measures of predictive ability. In cross valdation, predictive ability can be computed directly by comparing predicted and observed data in the validation set, e.g. via the correlation.

7. PLOS authors have the option to publish the peer review history of their article (what does this mean?). If published, this will include your full peer review and any attached files.

Reviewer #1: No

---

## [Author Response · Author response to Decision Letter 3]

26 Jan 2021

Reviewer #1: 

The authors assert that the deviance is better when P0 is imputed than when it is estimated. This suggests that something is wrong. The authors are using maximum likelihood estimation, so the value for P0 they find by that method should maximize the likelihood, and hence minimize the deviance. This can't possibly be beat by imputing P0 in any way. All of this is assuming, of course, that both times they are using the exact same data and the exact same likelihood function, which is necessary if you want to use deviance or AIC to compare models.

Response: Thanks for the reviewer’s comment. The values of P0 that used to establish the model were the actual CP rate at the first row of each furrow. Because, there were some missing values of the CP rate at the first row, the missing values were imputed by the k-NN algorithm for fitting the model. In this revision, we also provided the fitting result that P0 was defined as a parameter in the supporting information. Then, the process of the comparison of model was used the same likelihood function (Poisson likelihood function) and the same dataset.

The authors state that they fitted the models as generalized linear models (GLM) but the models as stated in equations (2) to (8) are not generalized linear models. Several of them can be turned into a GLM upon taking the logarithm, and I am assuming this is what the authors have done. This would need to be explained succinctly. The parameter P0 of the model CP4 will be an intercept, but on the log-scale. Note that models CP6 and CP7 are not GLMs so it is not clear what was done to fit these. Since the authors are using SAS, they could use the NLMIXED procedure to fit these model (and other models for comparison to make sure they get the likelihood right). At any rate, it would be good to tell readers how exactly the different models were fitted in SAS. 

Response: Thanks for the reviewer’s comment. All models have been taken the logarithm (the log link function) to get the linear predictor of models. In the condition that P0 is known, the P0s of the CP4 and CP7 were combined with the offset term of the models. The P0 was defined as the log-scale intercept in the fitting result of models with parameter P0. The SAS procedure used to fit the models was PROC GENMOD. The detail of model fitting has been added in the subsection of Statistical analysis.

One further issue: The authors assert that they assessed "predictive ability" by deviance and AIC. These are not measures of predictive ability. In cross validation, predictive ability can be computed directly by comparing predicted and observed data in the validation set, e.g. via the correlation.

Response: Thanks for the reviewer’s comment. In Fig. 5, the correlation coefficient between predicted and observed data was showed in the original validation data. We have added the correlation coefficient between predicted and observed data in the Table 5 for the simulated validation sets. Also, the description of deviance and AIC was corrected (lines 335, 346, 450-451, 456-457, 522, 531-532, 545-546).

---

## [Decision Letter · Decision Letter 4]

1 Feb 2021

PONE-D-20-23403R4

Bootstrap simulations for evaluating the model estimation of the extent of cross-pollination in maize at the field-scale level

PLOS ONE

Dear Dr. Lin,

Thank you for submitting your manuscript to PLOS ONE. After careful consideration, we feel that it has merit but does not fully meet PLOS ONE’s publication criteria as it currently stands. Therefore, we invite you to submit a revised version of the manuscript that addresses the points raised during the review process.

We look forward to receiving your revised manuscript.

Kind regards,

Mehdi Rahimi, Ph.D.

Academic Editor

PLOS ONE

Reviewers' comments:

Reviewer's Responses to Questions

**Comments to the Author**

1. If the authors have adequately addressed your comments raised in a previous round of review and you feel that this manuscript is now acceptable for publication, you may indicate that here to bypass the “Comments to the Author” section, enter your conflict of interest statement in the “Confidential to Editor” section, and submit your "Accept" recommendation.

Reviewer #1: (No Response)

2. Is the manuscript technically sound, and do the data support the conclusions?

Reviewer #1: (No Response)

3. Has the statistical analysis been performed appropriately and rigorously? 

Reviewer #1: (No Response)

4. Have the authors made all data underlying the findings in their manuscript fully available?

Reviewer #1: (No Response)

5. Is the manuscript presented in an intelligible fashion and written in standard English?

Reviewer #1: (No Response)

6. Review Comments to the Author

Reviewer #1: The authors maintain that P0 should be imputed and then treated as an offset in the ML estimation of the GLMs. I can only re-iterate that it is better to estimate P0 when fitting the GLM. The deviance and AIC will be better. Conversely, if they use P0 as an offset, they need to make sure the intercept is taken out of the model, using the NOINT option. As this is not mentioned in the paper, I am not sure the models were fitted correctly, even if one accepts that P0 should be an offset.

I do not think the authors have properly fitted model CP7 because there is no way to turn this into a GLM, yet the authors used the GENMOD procedure to fit it, and this is a procedure to fit GLMs. The authors asser that " In addition, model CP7 was taken the logarithm after subtracting 0.003." Have they substracted 0.003 from the observed response and then fitted a GLM? In that case, they have not fitted the model properly by ML. And the "data", i.e. the response, is not the same, so they cannot compare the deviances and AIC values with the other models. Again, I can just re-iterate my suggestion to fit the model properly using NLMIXED.

If the authors have not actually fitted models CP5 and CP6, but used fits from literature, how did they obtain deciance and AIC?

7. PLOS authors have the option to publish the peer review history of their article (what does this mean?). If published, this will include your full peer review and any attached files.

Reviewer #1: No

---

## [Author Response · Author response to Decision Letter 4]

24 Feb 2021

Reviewer #1:

The authors maintain that P0 should be imputed and then treated as an offset in the ML estimation of the GLMs. I can only re-iterate that it is better to estimate P0 when fitting the GLM. The deviance and AIC will be better. Conversely, if they use P0 as an offset, they need to make sure the intercept is taken out of the model, using the NOINT option. As this is not mentioned in the paper, I am not sure the models were fitted correctly, even if one accepts that P0 should be an offset.

Response: According to the Table S1 and Table 2, the models with P0 as an estimated parameter perform worse than models with imputed P0. The P0 was combined with the total grain number and was treated as an offset. The option NOINT was used.

I do not think the authors have properly fitted model CP7 because there is no way to turn this into a GLM, yet the authors used the GENMOD procedure to fit it, and this is a procedure to fit GLMs. The authors asser that " In addition, model CP7 was taken the logarithm after subtracting 0.003." Have they substracted 0.003 from the observed response and then fitted a GLM? In that case, they have not fitted the model properly by ML. And the "data", i.e. the response, is not the same, so they cannot compare the deviances and AIC values with the other models. Again, I can just re-iterate my suggestion to fit the model properly using NLMIXED.

Response: After subtracting 0.003 from the observed response, the CP7 can be transformed into GLM. Because the models don’t include the random effect, using NLMMIXED is not necessary. 

If the authors have not actually fitted models CP5 and CP6, but used fits from literature, how did they obtain deciance and AIC?

Response: With a proper likelihood function, the deviance and AIC can be calculated from any model with observations. The CP5 and CP6 was treated as a fitted model in our study.

---

## [Decision Letter · Decision Letter 5]

2 Mar 2021

PONE-D-20-23403R5

Bootstrap simulations for evaluating the model estimation of the extent of cross-pollination in maize at the field-scale level

PLOS ONE

Dear Dr. Lin,

Thank you for submitting your manuscript to PLOS ONE. After careful consideration, we feel that it has merit but does not fully meet PLOS ONE’s publication criteria as it currently stands. Therefore, we invite you to submit a revised version of the manuscript that addresses the points raised during the review process.

We look forward to receiving your revised manuscript.

Kind regards,

Mehdi Rahimi, Ph.D.

Academic Editor

PLOS ONE

Journal Requirements:

Reviewers' comments:

Reviewer's Responses to Questions

**Comments to the Author**

1. If the authors have adequately addressed your comments raised in a previous round of review and you feel that this manuscript is now acceptable for publication, you may indicate that here to bypass the “Comments to the Author” section, enter your conflict of interest statement in the “Confidential to Editor” section, and submit your "Accept" recommendation.

Reviewer #1: (No Response)

2. Is the manuscript technically sound, and do the data support the conclusions?

Reviewer #1: (No Response)

3. Has the statistical analysis been performed appropriately and rigorously? 

Reviewer #1: (No Response)

4. Have the authors made all data underlying the findings in their manuscript fully available?

Reviewer #1: (No Response)

5. Is the manuscript presented in an intelligible fashion and written in standard English?

Reviewer #1: (No Response)

6. Review Comments to the Author

Reviewer #1: I think the authors need to reconsider the ML theory underling GLMs. The way they responded to my comments indicates that they have not understood my points. I am a bit at a loss what else I can do to make them understand that their analysis is flawed regarding model CP7, and also regarding the estimation of P0 in some of the other models. I'll give it one last try. I'm really trying to help the authors and make sure they do not publish a flawed analysis.

(1) If you are using ML estimation properly, the deviance can only improve when including P0 in the estimation of the model, as compared to imputing P0 in any way. If you find the opposite, which seems to be the case, this is a sure sign that something is wrong in your analysis.

Subsequently, I am particularly refering to these two statements in the authors' response:

"Response: After subtracting 0.003 from the observed response, the CP7 can be

transformed into GLM. Because the models don’t include the random effect, using

NLMMIXED is not necessary."

"Response: With a proper likelihood function, the deviance and AIC can be calculated

from any model with observations. The CP5 and CP6 was treated as a fitted model in

our study."

(2) Just because you can compute a likelihood doesn't mean you can use it to compare model fits. You can only use the likelihood to compare model fits if the data used to fit the different models is exactly the same. So you can't substract 0.0003 from the response for one model but use the raw data "as is" for the others and then use the AIC or deviance to compare model fits. This is a fundamental fact in likelihood theory, and I would strongly urge the authors to consult their favourite mathematical statistics textbook on this point.

(3) If you substract an arbitrary value from a count, it's no longer a count and so it can't reasonably be assumed to follow a Poisson distribution after this operation. I can only re-iterate that the best to fit model CP7 is to use NMLIXED, leaving all model terms on the right-hand side.

7. PLOS authors have the option to publish the peer review history of their article (what does this mean?). If published, this will include your full peer review and any attached files.

Reviewer #1: No

---

## [Author Response · Author response to Decision Letter 5]

17 Mar 2021

Reviewer #1:I think the authors need to reconsider the ML theory underling GLMs. The way they responded to my comments indicates that they have not understood my points. I am a bit at a loss what else I can do to make them understand that their analysis is flawed regarding model CP7, and also regarding the estimation of P0 in some of the other models. I'll give it one last try. I'm really trying to help the authors and make sure they do not publish a flawed analysis.

(1) If you are using ML estimation properly, the deviance can only improve when including P0 in the estimation of the model, as compared to imputing P0 in any way. If you find the opposite, which seems to be the case, this is a sure sign that something is wrong in your analysis.

Subsequently, I am particularly refering to these two statements in the authors' response:

"Response: After subtracting 0.003 from the observed response, the CP7 can be

transformed into GLM. Because the models don’t include the random effect, using

NLMMIXED is not necessary."

"Response: With a proper likelihood function, the deviance and AIC can be calculated

from any model with observations. The CP5 and CP6 was treated as a fitted model in

our study."

(2) Just because you can compute a likelihood doesn't mean you can use it to compare model fits. You can only use the likelihood to compare model fits if the data used to fit the different models is exactly the same. So you can't substract 0.0003 from the response for one model but use the raw data "as is" for the others and then use the AIC or deviance to compare model fits. This is a fundamental fact in likelihood theory, and I would strongly urge the authors to consult their favourite mathematical statistics textbook on this point.

(3) If you substract an arbitrary value from a count, it's no longer a count and so it can't reasonably be assumed to follow a Poisson distribution after this operation. I can only re-iterate that the best to fit model CP7 is to use NMLIXED, leaving all model terms on the right-hand side.

Response: Thanks for the reviewer’s comment. In this edition, the models of CP5 and CP6 have been removed. The original model CP7 was renamed CP5 in this edition. All models were re-fitted by PROC NLMIXED with Poisson distribution and re-output the relative Tables and Figs in this revision.

---

## [Decision Letter · Decision Letter 6]

24 Mar 2021

Bootstrap simulations for evaluating the model estimation of the extent of cross-pollination in maize at the field-scale level

PONE-D-20-23403R6

Dear Dr. Lin,

We’re pleased to inform you that your manuscript has been judged scientifically suitable for publication and will be formally accepted for publication once it meets all outstanding technical requirements.

Kind regards,

Mehdi Rahimi, Ph.D.

Academic Editor

PLOS ONE

Additional Editor Comments (optional):

Reviewers' comments:

Reviewer's Responses to Questions

**Comments to the Author**

1. If the authors have adequately addressed your comments raised in a previous round of review and you feel that this manuscript is now acceptable for publication, you may indicate that here to bypass the “Comments to the Author” section, enter your conflict of interest statement in the “Confidential to Editor” section, and submit your "Accept" recommendation.

Reviewer #1: All comments have been addressed

2. Is the manuscript technically sound, and do the data support the conclusions?

Reviewer #1: Yes

3. Has the statistical analysis been performed appropriately and rigorously? 

Reviewer #1: Yes

4. Have the authors made all data underlying the findings in their manuscript fully available?

Reviewer #1: Yes

5. Is the manuscript presented in an intelligible fashion and written in standard English?

Reviewer #1: Yes

6. Review Comments to the Author

Reviewer #1: Great to see the full ML analysis of all models by NLMIXED, I think this rounds the paper off very nicely now.

The authors might consider adding standard errors for P0, a and b in Table 2.

7. PLOS authors have the option to publish the peer review history of their article (what does this mean?). If published, this will include your full peer review and any attached files.

Reviewer #1: No

---

## [Editor Report · Acceptance letter]

12 Apr 2021

PONE-D-20-23403R6 

Bootstrap simulations for evaluating the model estimation of the extent of cross-pollination in maize at the field-scale level 

Dear Dr. Lin:

I'm pleased to inform you that your manuscript has been deemed suitable for publication in PLOS ONE. Congratulations! Your manuscript is now with our production department. 

Kind regards, 

on behalf of

Dr. Mehdi Rahimi 

Academic Editor

PLOS ONE